# *MorphoNet*: an interactive online morphological browser to explore complex multi-scale data

Bruno Leggio[1,2,3], Julien Laussu[1], Axel Carlier[4], Christophe Godin[2,3], Patrick Lemaire [1,3] & Emmanuel Faure[1,2,3,4,5]

Powerful novel imaging and image-processing methods are revolutionizing many fields of biology, at scales ranging from the molecule to the functional organ. To support this big-data revolution, we develop a concept of generic web-based morphodynamic browser to interactively visualize complex image datasets, with applications in research and education. MorphoNet handles a broad range of natural or simulated morphological data, onto which quantitative geometric or genetic data can be projected.

[1] Centre de Recherche de Biologie cellulaire de Montpellier (CRBM), Univ Montpellier, CNRS, Montpellier 34293, France. [2] Laboratoire Reproduction et Développement des Plantes (LRDP), Univ Lyon, ENS de Lyon, UCB Lyon 1, CNRS, INRA, Inria, Lyon 69342, France. [3] Institut de Biologie Computationnelle (IBC), Univ Montpellier, Montpellier 34095, France. [4] Informatics Research Institute of Toulouse (IRIT), CNRS, INPT, ENSEEIHT, University of Toulouse I and III, Toulouse 31062, France. [5] Laboratoire d'Informatique de Robotique et de Microélectronique de Montpellier (LIRMM), Univ Montpellier, CNRS, Montpellier 34095, France. Correspondence and requests for materials should be addressed to E.F. (email: emmanuel.faure@lirmm.fr)

Over the past 20 years, the exploration of biological shapes across time, space, and scales has benefited from the emergence and refinement of novel imaging technologies[1], including IRM tomography, CT scans, cryoelectron microscopy and tomography, and light-sheet imaging[2]. These technologies, coupled with the development of fluorescent markers for cell compartments and of biosensors[1], reveal both the geometry of imaged specimens and their function. High-resolution data from living[3], fixed[4] or fossilized[5] material can now be produced, collectively bridging scales from the entire organism to single cells or even molecules, and revolutionizing fields as diverse as functional brain imaging and neuronal connectivity[6], paleontology[5] or morphogenesis[3]. In the field of developmental biology, the dynamics of the 3D shape, position, interaction and function of single cells, as well as cellular arrangements in different embryonic[7,8], larval[9] and adult[10] tissues are now accessible, redefining the horizons of modern morphometrics.

This burst of massive complex data brings forward the need for multidisciplinary approaches, unified formats and sharing environments for their quantitative exploitation. This situation shares numerous analogies with the next-generation sequencing revolution. Enormous amounts of genetic data were produced, which required the development of novel intuitive and interactive visualization tools, the web-based genomic browsers[11], to share, integrate and facilitate the human interpretation of genomic data across large scientific communities. Genomic browsers use genome assemblies as the basic framework onto which a variety of qualitative or quantitative pieces of information, at scales ranging from the single base to multi-megabase-sized chromosomal domains, can be projected.

There is currently no generic tool in the imaging field matching the power, generality and intuitiveness of genomic browsers. Existing morphological visualization platforms, either academic (e.g., OpenWorm[12], BodyParts3D[13], FPBioimage[14], Fiji[15], Icy[16], MorphoGraphX[17], and EmbryoMiner[18]) or commercial (Amira, Imaris, etc…) allow users to interact with specific datasets, within the context of precise scientific questions. Compared to genomic browsers, these platforms have three major limitations. First, they are not designed to integrate the user's data with external sources: none offers the possibility to project external quantitative/genetic information onto morphological datasets. Second, they are primarily designed as local analysis tools, and can hardly be used to share research data within and across scientific communities. Finally, the development of these tools requires the maintenance of a software which needs to be installed on different machines and with different operating systems who themselves evolve in time. This needs a long-term intensive plan of software maintenance, which may not be compatible with the limited resources available in the context of specific research projects, with the risk of rapid obsolescence.

Web-based morphological databases exist (MorphoSource, Phenome10k, MorphoMuseum, DataDryad), some of which come together with an online data visualization interface. However, the set of user interactions with the dataset is much more limited, and no additional information can be uploaded to and/or projected onto the dataset.

The open-source web-based tool presented here (http://www.morphonet.org) shows that the exploration and analysis of diverse large-scale imaging datasets can be used for a conceptually analogous philosophy to that which presided over the development of generic web-based genome browsers. MorphoNet allows the interactive visualization and sharing of complex morphodynamic datasets, onto which quantitative and qualitative information can be projected. Central to the concept is the definition of a unified, human-readable data format. In this sense, one could refer to MorphoNet as a morphodynamic browser.

## Results

**Conception of a morphodynamic web browser.** The first challenge was to compress massive fixed or live imaging datasets sufficiently to allow their visualization and interaction through the web. While raw microscopy images and their volume segmentations are too voluminous to be visualized online, segmented images are objects-based structures that can be highly compressed into surface meshes[19], without significant loss of shape information. MorphoNet then renders such collection of meshes using the Unity3D gaming engine, which runs on standard internet browsers and exploits the recent power of the WebGL. An intuitive, user-friendly web browser-based graphical user interface was developed to control the interaction with any dataset uploaded and stored in a common database (Fig. 1). To speed up rendering and visualization of cloud-stored datasets, including hundreds to thousands of objects, the level of details of the meshes can be set by the user. In such a way, complex 2D or 3D morphological datasets, at a single time point or across time series can easily be handled and shared. The whole open source code of MorphoNet and related documentation are available on a GitLab at https://gitlab.inria.fr/efaure/morpholab.

While the universal atomic unit in any genomic dataset is the DNA base pair, the atomic units vary between imaging datasets from complex organs, to cells or even subcellular structures or molecules. The choice of the scale of elementary building blocks for visualization, which depends on the scientific question, is thus left to the owner of the dataset. Figure 2 and Supplementary Fig. 1 illustrate that MorphoNet can render and interact in exactly the same way with any object-based meshed data obtained through microscopy imaging (Fig. 2a, f), computer simulation (Fig. 2b), X-ray tomography (Fig. 2c) or 3D drawing (Fig. 2d, e) across a broad range of scales of complexity.

In genomic browsers, bases are grouped into transcripts, genes or other genetic elements through the use of universally formatted files[11], that can be shared and compared within and between research networks. We likewise developed a strategy to describe biological structures as multiscale systems and to express this using hierarchies of topological and geometric objects, such as cellular complexes. First, in the case of multichannel image acquisition, MorphoNet can group and independently visualize the information present in the different channels, for example to visualize several organelles within each cell. Importantly, objects in different visualization channels can be linked and actions performed on one channels can be automatically reported on linked objects on other channels. Second, individual objects can also be grouped in space, for example to build higher-order structures, such as organs (Fig. 2d, e) or to identify all neighbors of an object of interest according to adjacency relationships (Supplementary Fig. 1, Supplementary Movie 1). In the case of time series, objects can finally be grouped by their temporal ancestry, for example to track cell lineages (Fig. 2a).

Individual objects, as well as spatial or temporal groups, can be identified by the assignment of specific color labels, which can be propagated through time series. This offers flexibility in observing and interacting with shape changes of objects, and in exploring the processes characterizing their morphological history (Fig. 2a, b, Supplementary Movie 2). Objects and groups can be hidden or made partially transparent, either at a single time-point or throughout time series. As shown in Fig. 2d, e, this feature allows the exploration of internal structures within highly complex multilayered datasets such as a whole human body (Supplementary Fig. 1, Supplementary Movies 3–6).

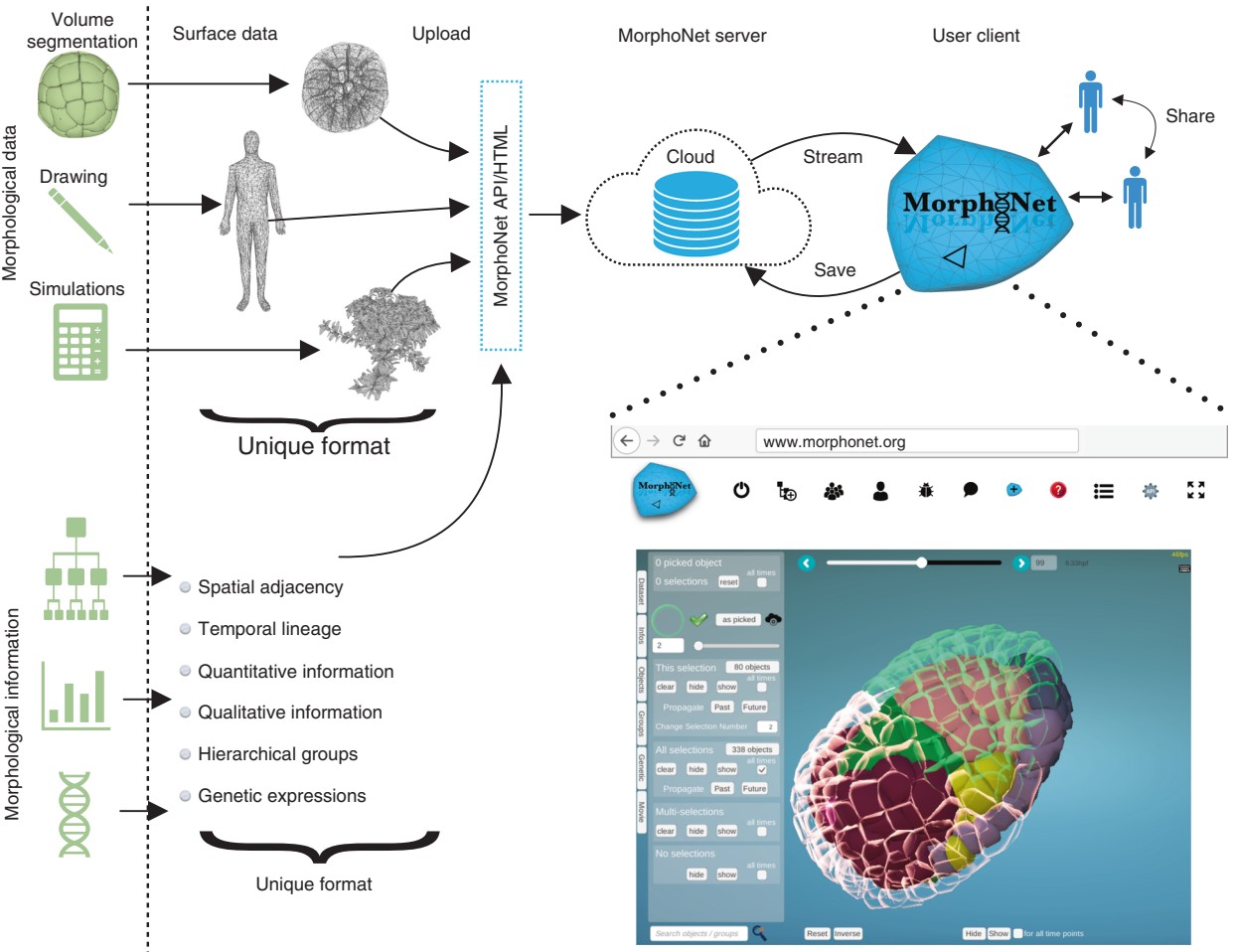

**Fig. 1** MorphoNet framework and its online graphical user interface. The cloud-based structure and the use of unifying formats for both morphological datasets and related information are the key concepts of our online morphological browser

To facilitate the interpretation of project-specific data in the context of publicly available information, genomic browsers support the upload by private users of custom datasets, such as RNA-seq or ChIP-seq, mapped onto the reference genome and which appear as individual genetic tracks. MorphoNet also supports the upload of formatted external quantitative and qualitative information, mapped onto individual objects, and which can be projected onto individual objects or groups in the form of color labels, text or heat maps (Fig. 2c, f, Supplementary Fig. 1, Supplementary Movies 2–10). Such morphological tracks can be saved in the cloud and shared with other users.

Among the possible morphological tracks, one of notable interest is gene expression patterns. MorphoNet offers the possibility to upload gene expression data from in situ hybridizations, cell- or tissue- specific RNAseq, Starr-seq, or chromatin signatures and to superimpose them onto a 2D or 3D scaffold as either boolean (color selections) or quantitative (heatmaps) information. This feature bridges cell-level genetic and 3D dynamic morphological information in one visualization tool, permitting the exploration of their spatio-temporal interplay (Fig. 3, Supplementary Movie 9). Genetic information can either be automatically synchronized from external databases (such as ANISEED[20]) connected to MorphoNet through their Application Programming Interfaces (APIs) (Fig. 3a) or uploaded manually by users in the dedicated genetic MorphoNet format, detailed in the section "Morphological information" in the Methods (Fig. 3b).

Finally, MorphoNet can also be used to explore population variabilities or evolutionary divergence. For instance, morphological or genetic variations within or between species or between wild-type and experimentally- or environmentally-manipulated samples can be explored in time and space through the upload of formatted comparative morphological or genetic information. Interspecies variability of relative tissue volume of developing embryos can, for instance, be pre-calculated and visualized as a heat map, to show whether it is localized on certain tissues and/or certain developmental stages. In addition averages, standard deviations, relative variations, and other relevant statistics can be calculated on quantitative properties.

A few simple calculations (average, standard deviations, normalizations) exist as built-in options in the MorphoNet interface. Due to the web-based structure of the tool, more costly calculations on the server would affect the user dynamic interaction with the dataset. However more complex calculations can be performed externally. The Morphonet Python API and ImageJ plugin provide an easy interface to upload any Python-based or ImageJ-performed calculations directly to the MorphoNet server. Otherwise, the user has always the possibility to create.txt files and upload them from the online interface.

**Potential use in research**. To exemplify the use, richness, and generality of MorphoNet we provide three detailed tutorials explaining how MorphoNet can address specific needs. These

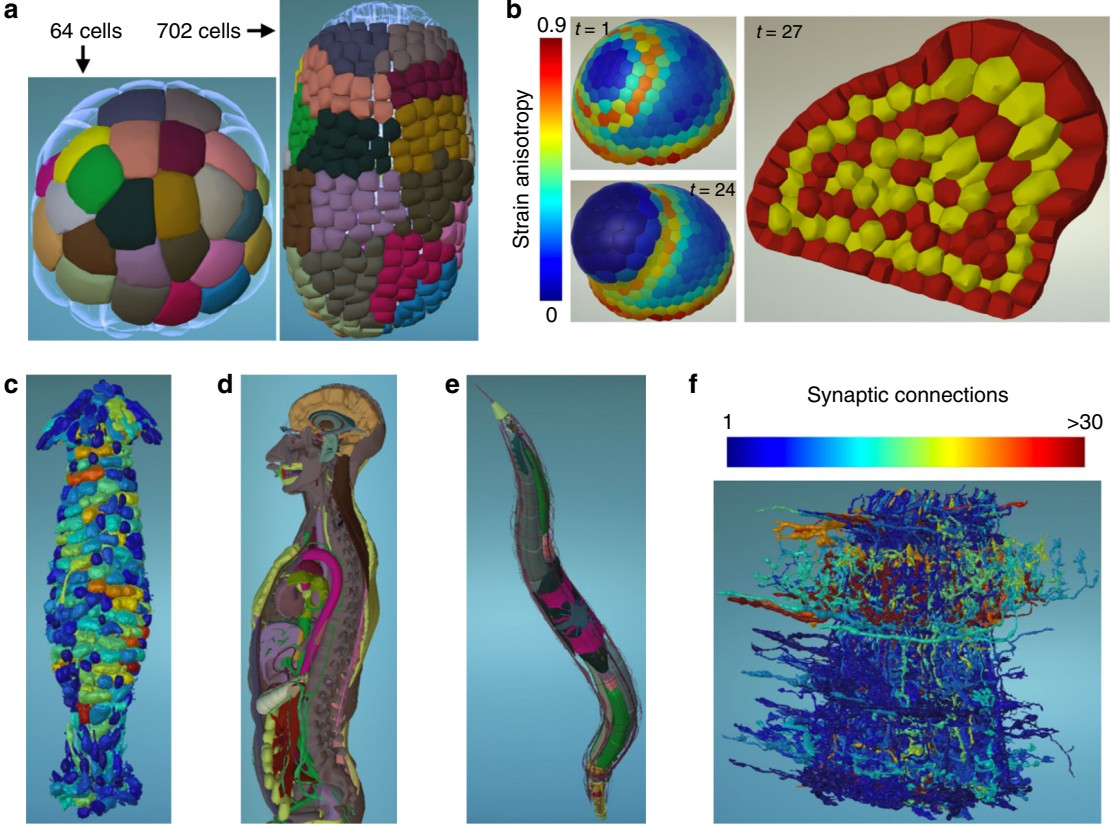

**Fig. 2** Morphological datasets and information. **a** Tracking of epidermal clones through several rounds of cell division in *Phallusia mammillata* embryo. Epidermal cells are color-coded at 64-cell stage and can hence be followed until 702-cell stage. Published data[7]. **b** Simulation of an abstract lateral organ formation at an oryzalin-treated shoot apical meristem. Growth is simulated according to Boudon2015[21], with the wall elasticity model described in Oliveri2018[22]. Strain anisotropy during growth is shown as heatmap for two time points on the left, and inner cell layers are highlighted in alternating red-yellow colors on the right. Data kindly provided by H. Oliveri. **c** 3D structure of a nest of termite *Cubitermes*[23], in which a heat map has been applied to each chamber showing the chambers' volume distribution (red = high volume, blue = low volume). Data kindly provided by A. Perna. **d** Exploration of a human body atlas, in which color labels are applied corresponding to different classes of organs. Dataset can be cut to easily explore its inner structure. **e** Inner structure of *C. elegans* worm. To reveal inner organs, hypodermis layers were hidden from view. Color labeling identifies different organ groups. **f** Quantitative information visualization by heat map on Drosophila connectome[6], representing the distribution of number of synaptic connections for each axon. More information about these datasets can be found in the Supplementary Notes section of the Supplementary Information

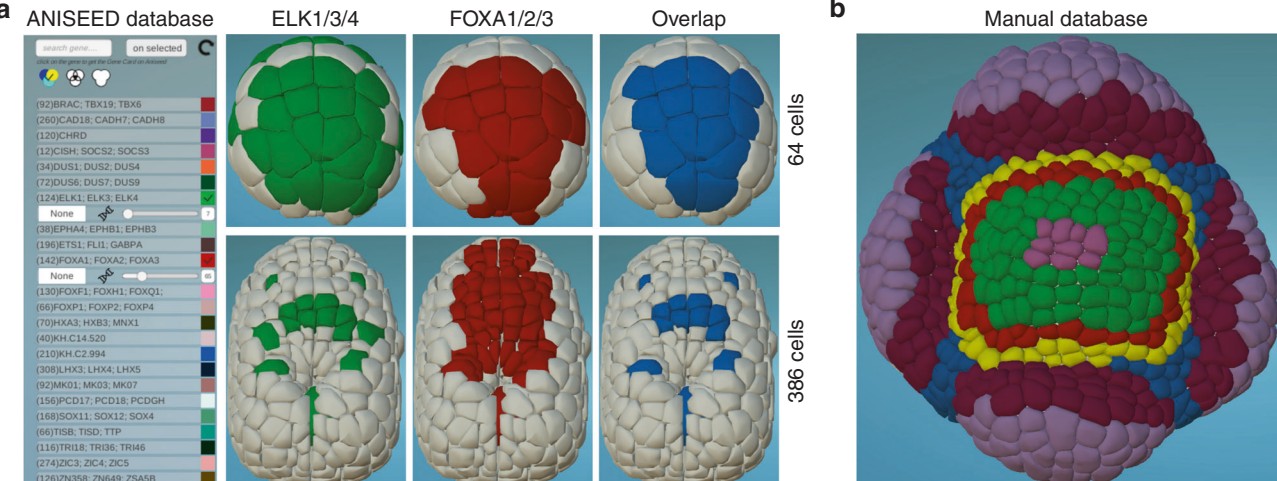

**Fig. 3** Genetic Information. **a** Gene expression profiles on ascidian embryos. The panel shows the genetic interface, with the list of ascidian genes taken directly from the online database ANISEED[20]. Two expression profiles are shown in green and red at two different embryonic developmental stages, together with their overlap in blue. Data taken from ref. [7]. **b** Gene expression in *Arabidopsis* meristem. The panel shows expression domain overlaps of different genes, taken from a manually entered formatted file. Each color corresponds to a different set of genes expressed in the territory. Data kindly provided by J. Traas (unpublished)

tutorials can be found online at http://www.morphonet.org/Tutorials/MorphoNet_Tutorials.php.

The first tutorial illustrates how to use MorphoNet to investigate dynamic biological properties of embryonic cells for the specific case of ascidian embryonic development. Thanks to its straightforward upload process, a 3D reconstruction is available on the MorphoNet server as a new dataset immediately after segmentation. MorphoNet is then used to visualize complex 3D dynamic shape changes and to select, on 3D data and though the timestack, a group of cells of interest to study (specifically, those cells whose shape changes the most during gastrulation, i.e., endodermal cells). A few lines in Python are then used to quantify the change in time of endodermal cells' shape during gastrulation. These results are then uploaded to MorphoNet in the form of quantitative information. MorphoNet allows to project this dynamic information on the embryonic cellular scaffold and to visualize its evolution. In this way, by a simple and direct visual investigation, one can clearly see the onset of the two main shape changes involved in ascidian gastrulation, and confirm their sequentiality in time.

The second tutorial shows how to use the MorphoNet plugin for ImageJ/FiJi we have developed. It allows to use the many image analysis tools, including properties quantification, available in ImageJ and to upload the results directly to the MorphoNet cloud. Specifically, the tutorial shows how one can import a volumetric segmented image of one ascidian embryo in ImageJ, create a surface mesh with the Fiji built-in library and upload it as a new dataset to MorphoNet, together with the quantitative information about the volume of each cell in the embryo.

The third tutorial demonstrates the use of MorphoNet to help a 3D+t segmentation and cell tracking routine. It illustrates how MorphoNet and its functions can be used to correct some of the most common problems occurring during the segmentation of volumetric images, distributed in timestacks: over-segmentations of cells, under-segmentations of cells and missed temporal links between mother cells and daughter cells after cell divisions.

## Discussion

In summary, MorphoNet integrates, in one open-source online tool (http://www.morphonet.org), a comprehensive palette of interactions to explore the structure, dynamics and, variability of biological shapes and to confront these with genetic information. It largely profits from the most recent and well-developed web technologies and introduces a generic strategy for hierarchical representation of biological structures. We expect in the future, thanks to the contribution of large user communities, to develop the MorphoNet data and metadata formats towards standardization.

Due to the lack of unified formats for morphological data and metadata, uploading a dataset from an external database in Morphonet leads to the duplication of this dataset in a slightly different format. To reduce the impact of this reaccessionning issue, we invite Morphonet users to always provide the original dataset ID in the metadata of the uploaded copy, and to cite both MorphoNet and the original database and ID in scientific publications.

MorphoNet allows the efficient and controlled dissemination and sharing within scientific communities of public and private information, with large visibility for new discoveries. It fills a gap which has until now limited the quantitative understanding of morphodynamics and its genetic underpinnings by contributing to the creation of ever-growing atlases of morphogenesis. In addition to its research use, Morphonet is expected to give access to the public and educational professionals to the immense body of live imaging data that is increasingly made publicly available by academic institutions on open-access data repositories, thereby facilitating the dissemination of scientific knowledge across society.

### Table 1 MorphoNet user rights

|  | Creator | Owner | Reader |
|---|---|---|---|
| Can visualize meshes | X | X | X |
| Can see shared Information | X | X | X |
| Can see and delete private information | X |  |  |
| Can upload/download information | X | X |  |
| Can download meshes | X | X |  |
| Can share information | X | X |  |
| Can grant access rights | X | X |  |
| Can erase dataset | X |  |  |

## Methods

**MorphoNet features and functions**. MorphoNet has a highly interactive interface through which users can visualize morphodynamic data and display quantitative and qualitative information.

The first public instantiation of MorphoNet is currently hosted on a secure server (http://www.morphonet.org). The system is portable and, as the bulk of the calculations is handled on the client side, it can be installed on any server with even modest computing power.

MorphoNet has two different types of access possibilities: registered users, whose access rights are detailed in Table 1, and non-registered users, who can access public datasets and have visualization rights only.

A visual help web page is accessible at any time through the question mark button at the top of MorphoNet page. Supplementary Movies 1–10 provide several practical examples of use of the MorphoNet interface.

MorphoNet offers a large variety of interactive features to explore the structure and dynamics of morphological datasets. Basic dataset interactions include data rotation and translation by keyboard + mouse shortcuts, object selection by mouse clicks or keyboard + mouse shortcuts, zooming in/out through mouse actions and scrolling through timestacks by either a scrollbar or by keyboard shortcuts. All keyboard and mouse actions are explained in the navigation menu in the upper-left corner of the application screen.

In addition to these basic features, more advanced data interactions are available. These functions are grouped into 6 menus in the visual interface: Dataset, Infos, Objects, Groups, Genetic, Movie. Moreover, standard object-interaction functions are given at the bottom of the interface.

Standard object-interaction functions allow users to search specific objects or groups of objects, identified by their name or numerical ID. Once objects are picked, they can be hidden from view, either at specific time instants or throughout the timestack. User can also select all physical neighbors of any picked objects, which can be iteratively repeated to explore the structure of object adjacency.

In the Dataset menu, users can interact with the dataset as a whole. The dataset position and orientation in space can be reset to default values, and the default orientation can be set (only by dataset owner).

Dataset can be cropped along the three main axes, or along a custom direction. Users can also switch among different visualization channels, or download surface meshes or visualize adjacency and temporal lineage trees. In particular, nodes in the lineage tree are automatically linked to their corresponding objects in the dataset: any action performed on objects (such as color-labeling) is automatically reported on tree nodes.

Infos menu lists all information associated with the dataset. Information is categorized as: Selection, Quantitative, Qualitative and Genetic. Selections are color labels applied to objects (see later on about the Objects menu). Quantitative information, associated with all objects or to a selected subgroup, can be visualized as heat maps on the dataset, and its numerical value is shown for any picked object. In addition, users can manipulate quantitative information by, for instance, calculating time and space averages, mean values, standard deviations or time normalization. After such manipulation, a new entry in the Quantitative category is created, representing the outcome of the requested calculation.

Qualitative information associates a string of characters to all objects or a selected subgroup. Once one Qualitative entry is activated, the corresponding information is visualized as text superimposed on the dataset as the mouse pointer passes over the corresponding object.

Genetic information associates objects to an expression level of a specific gene. This information replaces or complements the equivalent information extracted from external genetic databases (if connected to MorphoNet). Expression level can be given as boolean (on/off) or as quantitative level. If one or more Genetic entries are activated, a Genetic menu is created and made accessible (see below).

From the Infos menu, users can also upload to the cloud text files (.txt extension) containing new information. Users also have the possibility to manually create and curate and upload new information tracks directly through the visual

online interface. All the information provided can be downloaded as text files, shared with selected users, and manually curated.

Specific object interactions are possible in Objects menu. There users can apply color selections to picked objects, hide non-selected objects from view and propagate color-labeling through time along the lineage tree. This allows users to track the evolution of selected objects in time and/or to cluster groups of objects in space.

Groups gives a list of previously uploaded sets of objects of particular relevance (these can, for instance, be tissues in organs). The group structure is hierarchical, such that smaller groups can build up larger ones, highlighting the relevant morphological scales of the dataset. Users can select all objects (or subgroups) in one group and automatically apply color selections to each of them.

Genetic menu is present only if genetic expression data are provided, either by connecting MorphoNet to an external genetic database, or by uploading it through the specific genetic information format, or by a combination of the two.

The menu shows the full provided list of genes, with the gene name, the total number of objects expressing the specific gene and the possibility to color-label these objects. In addition to single gene expressions, users can also visualize using a color-code the overlap and the union of expression patterns of two or more genes. If quantitative expression levels are provided, users can also visualize them as heat maps. Users can in addition obtain the list of genes expressed by any picked object, or search specific genes by name.

Genetic expression patterns can also be manually curated directly from the Genetic menu.

Screenshots of visualized dataset, as well as movies of spatial and temporal dynamics, can be produced in the Movie menu.

When creating a movie (Scenario), users can create keyframes of specific views of the data (both in space and in time) and can tune the frame-rate between two subsequent keyframes. In addition, they can choose whether to interpolate dataset movements between two subsequent selected views. Once a movie has been edited, users can visualize it and download it as an.mp4 file. Movies can also be uploaded to the cloud and shared with other users.

For both screenshots and movies, all user interactions with objects (e.g., color selections) are taken into account.

Finally, depending on the user account level and on the user rights on each specific dataset, user account options may allow users to modify and erase dataset information, share the whole dataset and, eventually, its information with other specific users, change the default point of view or erase the whole dataset.

Users can always update their personal information and send feedback/comments to administrators.

**MorphoNet implementation**. MorphoNet is a web-based application composed of a front-end application on the client (internet browser) and a back-end support on the server which communicate through classical http requests.

The structure of MorphoNet integrates a database for uploading and sharing, which is hosted on the back-end, and a 3D interactive interface for visualization and other advanced dataset interaction, which is hosted on the front-end.

The front-end application of MorphoNet is accessible through the url http://www.morphonet.org on any Internet browser, but we strongly recommend to use Firefox, which is free and accessible on any operating system. We also extensively tested MorphoNet on two versions of Chrome (72.0.3626.121 and 73.0.3683.75) on Windows 7, 8, 10, on MacOS Sierra 10.13, Mojave 10.14 and on Ubuntu 14, 16, 18. MorphoNet is fully compatible with the tested Chrome browser versions.

MorphoNet uses the browser internal library WebGL 2.0 to display complex morphological data, and thus needs a recent computer with enough memory and CPU power to support this library. The main technical limitation of MorphoNet, imposed by the very structure of WebGL, is the 2 GB limit of memory use, as is the case for many online mesh visualization platforms. However, for a better interactivity with the dataset during more advanced user interactions (information upload, computations, lineage tree exploration), we suggest to limit the total size of a dataset mesh to 500 MB. This can be achieved by decimating the dataset mesh, i.e., decreasing the number of polygons composing the mesh, for instance through MeshLab (http://www.meshlab.net).

In the same spirit, in order to optimize the interaction with the dataset, we recommend to limit the total number of elementary objects in each dataset to half a million. Also, for optimal loading, the dataset should not contain more than 1 thousand.obj files (each.obj file can contain more than one elementary object, see "Format specification" section).

The front-end application is developed in Unity 3D v2017 Personal Edition using C# language. During the building of the application, Unity converts all code to JavaScript, which makes it readable by any browser.

We provide, by application/request, the possibility to add new data on our own server. A database implemented in MySQL stores all necessary information of MorphoNet (datasets, users, quantitative data, curations, etc.). The server uses classical apache and PHP libraries to communicate with the browser. Any researcher/laboratory can also install MorphoNet on their own server to create their own private morpho-browser.

Researchers with little computer science knowledge can create a new dataset and upload their meshes directly through the web interface (as.obj files, see "Format specification" section). For a more controlled upload, however, we

recommend to employ the MorphoNet API written in Python, which is provided at the url http://www.morphonet.org/HELP/HelpAPI.php. The API allows uploading meshes distributed in multiples files, their associated information, and permits to perform common user functions (e.g., create, share or delete datasets).

Once data are uploaded to the server, users can directly visualize and interact with them on MorphoNet. For a smoother usage, a daemon, running on the server, converts the.obj format of each new mesh in AssetBundle, which is the optimized Unity format.

**User rights: creation of users and groups**. Any registered user on MorphoNet has the right to create new users. Once created, a user can be erased only by him/herself or by MorphoNet administrators.

Any registered user on MorphoNet has the right to create a user group. Groups are used to easily share datasets/information with all group members.

These groups can be either private or public: in private groups, only group managers can add new members, while any users can freely decide to join a public group.

The group creator is automatically a manager. Other group members can be either managers or simple members. Both managers and members have access to group datasets and information, while only managers can add/remove users (for private groups), manage the group structure and allow manager rights to other members.

Once created, groups can be completely erased from the database of MorphoNet by MorphoNet administrators only.

Group managers can, however, decide at any time to make the group inactive by archiving it. The group then becomes hidden and its rights to dataset access are suspended. Managers can reactivate archived groups at any time.

**User rights: dataset and information access and sharing**. Once a dataset is created by uploading surface meshes to the MorphoNet database, it will stay private until the creator of the dataset decides to give access to other selected users (or user groups).

As such, each dataset has a community of allowed users who can, with different access rights, read and/or upload information to it.

Allowed users for each dataset in MorphoNet can be granted three access-rights levels:

Creator, i.e., the user who uploaded the meshes to the cloud; there is one Creator only per dataset;

Owner, i.e., users having full-right access to meshes and information; each dataset can have several Owners;

Reader, i.e., users having read-only access to meshes and information; each dataset can have several Readers.

Users who have not been given access to a specific dataset will neither be able to see it nor to access it.

Both Creator and Owners can upload new information to the dataset. All uploaded information can either be kept private, or shared will all users with read access (Creator, Owners, and Readers).

Creator and Owners can also download both meshes and all information they have access to.

Private information can only be seen by the specific Owner who uploaded it and by the Creator of the dataset, who has full access to everything associated with his/her dataset.

Both the Creator and Owners can grant access rights to other users or user groups.

Readers have no upload/download/share right and can only visualize meshes and shared information.

A dataset can also be made public, in which case all registered users on MorphoNet automatically acquire Reader rights.

In addition, public datasets can be accessed by users from anywhere in the world without the need for a MorphoNet account: public access will be granted to unregistered users, with visualization rights only.

**Format specification**. In order to fully exploit the interdisciplinary networking structure of the idea behind morphological browsers we have hereby introduced, we created a generic format for morphological information specification.

To this end, we take advantage of the object-based structure of morphological datasets by introducing morphodynamic information categories which can be projected on elementary blocks. Each of these objects can be uniquely identified by an object tuple (OTP) given by its context. For example, when both temporal information and several visualization channels are present in a dataset, at any given time point and for any given visualization channel, one is left with a collection of objects. Therefore, OTPs in such datasets correspond to three integer numbers separated by commas: t, id, ch, specifying the time point (t), the visualization channel (ch) and the ID of the specific object given in the obj file (id).

On the other hand, if only one time point and one channel is provided in the dataset, OTPs can be simplified to just the simple *id* specified in.obj files.

**Morphological datasets**. Morphological datasets must be uploaded to MorphoNet in the form of surface meshes. More specifically, we employ one of the most used

### Table 2 Overview of morphological information formats

|  | Information type | Information format |
|---|---|---|
| Temporal information | Time | OTP1: OTP2 |
| Spatial information | Space | OTP1: OTP2 |
| Group information | Group | OTP: group1 (: subgroup2...) |
| Quantitative information | Float | OTP: number (float) |
| Selection information | Selection | OTP: number (integer) |
| Color information | Color | OTP: R,G,B |
| Qualitative information | String | OTP: text |
| Genetic information | Genetic | OTP: number (float) |
| Dictionary information | Dict | OTP1: OTP2: number (float) |
| Sphere information | Sphere | OTP: x,y,z,r |
| Vector information | Vector | OTP: x1,y1,z1,s1: x2,y2,z2,s2 |

Further details are given in the text

format to represent meshes, i.e., the OBJ format. Each.obj file contains a list of elementary objects present at a given time point, of its vertices and of its faces. Each elementary object is identified by the time index at which it exists and its specific *id* at each given time, followed by the list of coordinates of its vertices and the list of faces composed of three vertices (each vertex being numbered starting from 1, with a global numbering running over all vertices of all objects at any given time point).

This leaves dataset owners the freedom to choose the level of detail of their morphological unit, which will then be interpreted by MorphoNet as the elementary object with which users can interact.

Each elementary brick should be referenced in the.obj file by the letter g (standard in the obj format) followed by its OTP.

For an optimal experience when interacting with the dataset, the total number of objects should not be higher than half a million. This is due to the intrinsic limits to the WebGL memory as of today.

**Morphological information**. Thanks to flexible OTPs, users can upload information to be visualized in the dataset. The general structure of information is a property associated with a specific object. The unique MorphoNet information format consists of a list of entries of the form OTP: property.

Properties can, in turn, represent several different kinds of morphological information, each identified by a specific information type.

The type of information must be mentioned at the beginning of the list as type: information type.

Each property must be given a name in order to uniquely identify it in the database.

Available information types and their format are listed in Table 2.

Specifically: Temporal information provides information on temporal relations between two objects OTP1 and OTP2 at different time points: OTP2 originates from OTP1 at a previous time point. This format can be used to identify the same object at two different time points and to characterize object divisions (for instance, if OTP1 divides and creates OTP2 and OTP3, the list will have both entries *OTP1: OTP2* and *OTP1: OTP3*) or objects fusion (if OTP1 and OTP2 merge to produce OTP3, the list will have both entries *OTP1: OTP3* and *OTP2: OTP3*).

Spatial Information provides information on spatial relations between two objects OTP1 and OTP2. It can be used to provide objects adjacency information or any other kind of connection existing between objects at the same time point. For instance, if one wants to provide the list of physical neighbors of object OTP1, the list will contain several entries *OTP1: OTP2, OTP1: OTP3*, etc…

Group Information provides group information on object OTP, used to identify several objects belonging to the same class (for instance, cells belonging to the same tissue). This information is given as text corresponding to the name of each group. Users can additionally provide subgroups information: their names must be given in a hierarchical order, from the largest to the smallest (i.e., subgroup2 is a part of group1 and so forth).

For instance, if OTP belongs to the group Epidermis and to subgroup Head Epidermis, the corresponding list entry will be *OTP: Epidermis: Head Epidermis*.

Quantitative Information provides quantitative information associated with object OTP. This information is a number, representing a certain property of OTP, such as for instance its volume. Such information will be displayed on MorphoNet as a heat map.

Selection information associates object OTP to standard MorphoNet selection, identified by an integer between 1 and 255 and corresponding to the selection number in the Objects menu of MorphoNet. Object OTP will then be color-labeled correspondingly.

Color information colors OTP according to given Red-Green-Blue values (integer in the range 0,255).

Qualitative Information provides qualitative information associated with object OTP. This information is in the form of text, representing a certain characteristic of OTP, such as for instance its name. Such information will be displayed on MorphoNet as text superimposed to OTP as the mouse pointer passes over it.

Genetic Information provides gene expression information associated with object OTP. The number characterizes the expression level (either given as boolean 0/1 or as more precise quantitative expression level). The gene name whose expression is given in the information must be specified as the name of the information file uploaded to the database.

Dictionary Information links object OTP1 to object OTP2 with a quantitative information on this link. Such a quantitative information is represented as heat map on the dataset. For instance, object OTP1 is associated with its direct physical neighbors, with additional quantitative information on their contact areas.

Sphere Information visualizes a sphere on the dataset, having the same color selection as object OTP, centered at coordinates x,y,z and with radius r.

Vector Information visualizes a line segment on the dataset, having the same color selection as OTP, starting at coordinates x1,y1,z1 with size s1 and ending at coordinates x2,y2,z2 with size s2.

**Reporting summary**. Further information on research design is available in the Nature Research Reporting Summary linked to this article.

## Data availability
All public datasets shown in this work are accessible through the MorphoNet online interface and available for download upon request.

## Code availability
The whole open source code of MorphoNet, its APIs, format converters, data analysis scripts in Python and related documentation are available on a GitLab at https://gitlab.inria.fr/efaure/morpholab.

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

## Acknowledgements

We thank M. Brozovic for helping with the MorphoNet-ANISEED connection. We thank J. Dardaillon for her precious help in the MorphoNet-ANISEED connection and in the NCBI taxonomy API. We thank F. Boudon for generously providing the dataset of simulated Mango tree. We thank J. Traas for kindly providing the dataset of *Arabidopsis* meristem and gene expression patterns, and for discussions and feedbacks. We thank A. Perna for providing the dataset of *Cubitermes* nest structure. We thank G. Cerutti for his help with the creation of surface meshes on volume segmentation datasets. We thank H. Oliveri for providing the dataset of simulated meristem. We thank C. Dantec and the CRBM for helping with and hosting the MorphoNet website. B. L., J. L., C.G., and P.L. acknowledge financial support from the Agence Nationale de la Recherche (Dig-Em contract, ANR-14-CE11-0013; Equipex Morphoscope2, ANR-11-EQPX-0029; Institut de Biologie Computationnelle, IBC, ANR 11-BINF-0002) and from the Inria Project Lab Morphogenetics.

## Author contributions

E.F. conceived the concept of MorphoNet, developed the code, contributed to its features and to the preparation of datasets and contributed to writing the manuscript; B.L. contributed to the development of the concept, the features of MorphoNet, the preparation of datasets and wrote the manuscript; P.L. contributed to the inception of the main concept, the development of features and of datasets and contributed to writing the manuscript; C.G. contributed to the inception of the main concept, the development of features and of datasets; J.L. contributed to the features and the preparation of datasets; A.C. contributed to development of concepts and features. All authors contributed to discussions on the structure of the manuscript.

## Additional information

**Competing interests:** The authors declare no competing interests.

