## [Peer Review File · Nature Communications]

Reviewers' comments:

Reviewer #1 (Remarks to the Author):

It's a great idea to have an online browser of 3D and 4D image data to allow researchers to share and interrogate data in an interactive way. This work attempts to provide this for segmented images. I think the authors should be commended for attacking this issue. The manuscript is clearly written and makes a strong argument for why such a facility is required.

I found using MorphoNet reasonably straightforward. The best data available to reviewers to try this out is currently the Phallusia embryo and it was relatively easy to get an idea of the basics of the morphogenesis of this simple embryo. Using the browser would be improved if combinations of the main dropdown menus were simultaneously available – for example I want to interact with the plane crops available under the Dataset dropdown while simultaneously viewing the list of genes whose expression profiles I had selected in the Genetic dropdown.

It would be good to see how successful this tool is when the embryo or organ is more complex. The zebrafish embryo example also stops at a relatively simple stage of development and the Drosophila embryo isn't a complete embryo so it impossible to gauge how this will deal with more sophisticated data. Can the authors say what the maximum number of cells and timepoints that could be handled by MorphoNet is? They do give some suggested maxima for file size and elementary objects but these are a bit abstract to me (someone that doesn't use segmented data), how does that translate to cells and timepoints?

I also tried out the Human brain anatomy and abdomen datasets as I teach these things to undergrads. I can't say I found these very useful. There are other tools and datasets available that are better (as least from the teaching point of view).

Of course the information available in these segmented images will only ever be as accurate as the segmentation of raw data and the annotation made by the uploading lab. So it's not really possible to know how much confidence to put on the information that you are trying to interrogate. Without the raw data or at least some estimate of the limitations/accuracy of the segmentation this will be a worry.

Will MorphoNet be a success? There's probably only one way to find out and that's to make it widely available and then monitor how many labs upload their data and how many other people successfully interrogate the shared data. I think its definitely worth finding this out for MorphoNet.

Reviewer #2 (Remarks to the Author):

In this paper Leggio et al. present a very innovative online computational tool -called MorphoNet- to explore complex dynamic morphological datasets using user-friendly interfaces with intuitive representation of biological entities, across organizational scales.

This project was inspired by the emergence of large scale datasets from state-of-the-art imaging technologies and the analogy with the concept of genome browsers, which offer a unifying platform to navigate other complex, sequence-based, datasets.

In this regard, the creation of MorphoNet is timely, and its potential impact could hardly be overstated.

Among the specific merits of the MorphoNet platform and the manuscript, the authors provide very strong justifications for a web-based analog to genome browser for morphodynamic data; they used clever formatting of segmented image datasets to allow portability; the tool allows for biologically relevant hierarchical groupings, and for computational integration of other datasets (such as sequencing data) onto a morphodynamic reference; finally, MorphoNet provides powerful visualization tools and even analytic functionalities.

I certainly commend the authors for this ambitious and important work, but nevertheless wish to provide some feedbacks, which I aim to be constructive, especially regarding the current format of

the paper.

GENERAL CONCERNS

First and foremost, the paper is well written, but from a very general perspective, which makes the specific aspects of MorphoNet sometimes hard to grasp. It almost reads more like an “infomercial” than an actual scientific article, and it is not entirely clear how one would use MorphoNet to address specific problems. Arguably, the tool’s potential is too vast to illustrate comprehensively, but perhaps using MorphoNet to address a particular set of biological problems, while navigating and introducing the various features of the software would better demonstrate the power of the tool.

Along the same line, the methods are written as a tutorial, but a verbose one, without the specific indications on how to perform indicated tasks. Best would be to have a proper methods section describing the main methods used to build MorphoNet, and then write a detailed tutorial to publish as supplemental information for prospective users. Here too, illustrating with specific scenarios would probably be more helpful than general statements such as “User can also [do this or do that]” without clear indication about how to.

SPECIFIC CONCERNS

It is not clear in the main text how the data used to create the reference models were collected, and processed. Although this information is available deep into the “METHODS” section, this could be addressed by providing very specific examples of using MorphoNet to address biological problems of interest.

Figure 3 shows gene expression patterns -presumably from in situ hybridization data- mapped onto embryo models, and the text mentions RNA-seq data, which is inherently more quantitative. I was wondering whether the latter can be represented as heatmaps to convey the expression level as well as the pattern, only to find answers deep in the “METHODS” section.

A more conceptual concern refers to the handling of perturbation datasets, considering that experimental perturbations may alter the morphodynamics used as reference. Here too, a test-case would better illustrate how MorphoNet can be used to integrate both the molecular and morphodynamic effects of perturbations than the general statements in the text.

Along the same line, it is not clear at all how MorphoNet can be used to compare morphodynamic processes across species. What would the morphodynamic reference be for different species? Here too, specific examples would be helpful.

The last part before the summary (LL 133-137) is important but explained very succinctly. Are the calculations for data analyses performed within MorphoNet? Or is it interfaced with popular data analysis software like Python or R? It might be useful to explain how MorphoNet can be integrated with these other popular softwares (e.g. Image J/Fiji “upstream” of MorphoNet; Python and R “downstream” of MorphoNet).

Similarly, it is not entirely clear from the paper how large genomics datasets would be integrated with morphodynamic data using MorphoNet.

MINOR CONCERNS

Abstract: suggest “research and education” instead of “research and teaching”

Reviewer #3 (Remarks to the Author):

In the manuscript “MorphoNet: An interactive online morphological browser to explore complex multi-scale data”, Leggio et al introduce a web-based tool to interactively visualize complex image datasets in a form of surface mesh with additional morphological information file as the input. They would like to develop a generic tool matching the generality and intuitiveness of genomic browsers.

Overall, online tool is a great approach especially with sharing property. Developing such online tool may meet many challenges. I think with the prepared data and external information matching the data, this is a very useful tool for teaching and presenting the data. But in the manuscript, there is no strong evidence to show how this tool has biological meaning in research. For example, there is no example to show any new biological findings by using this tool. There do exist some tools not only can visualize data but also can process and measure the data (e.g. drishti).

Meanwhile, with many constraints with the input files (e.g. surface mesh only, file size limitation, limited number of objects, morphological information has to be extracted by other imaging tools) and relatively simple properties, I think the current version is still preliminary and not generalized enough to satisfy broader users. I believe users do expect the online tool could match the local tool in terms of functions and properties.

Below, comments are more details.

- I understand that using only surface mesh is to save memory and to speed up. Only visualizing the surface mesh of a single object is not novel enough as too many tools can do it. To take advantage of this tool, the input data requires the volume segmentation first. For example, with the CT scans, the segmentation has to be done by other software with the volume data as input. After segmentation, different parts/objects will be labeled differently in a volumetric format. Many software such as drishti can visualize it in a similar way, and also can render and measure this volumetric data. But to be able to use MorphoNet, the users have to extract the surface mesh for each part from the volumetric data, assign labels, and then format it in a certain way for the input file of MorphoNet. In other work, it needs a lot of additional data processing to prepare the input files. If other tools can visualize it similarly without further data processing, the users may need stronger reasons to use MorphoNet.

- In the manuscript, the authors point out there are three major limitations for other existing morphological visualization platforms. However, I only partially agree with this current version of MorphoNet.

- 1) "...not designed to integrate the user's data with external sources" (line 48). I agree that the highlight of MorphoNet is that it can show the group information with those external information as input. It can hide and show any selected objects. The main concern is the external morphological information needs to be extracted by other tools and also needs to be formatted in a required way. This might limit the generalization. Many tools that can extract the morphological information also can visualize the data in a similar way.

- 2) "...local analysis tools can hardly be used to share research data" (line 50). Being able to share the data is a great idea, but I realize after interactively visualizing the data, the current version cannot save it. For example, what if the user would like to share the file after selecting and coloring. The current version cannot do it. In other words, the file is just shared as the file. Thus the tool is functioning as a drive or cloud similar to Google Drive and Dropbox. I think it is good to be able to save the script of what the user did and share it so that other users in the group could run and see it.

- 3) "...does not usually include a long-term plan of software maintenance" (line 52). I don't agree with it. I believe the maintenance of all the tools depends on the grant. Many tools keep updating for a higher version until today.

- The functions or properties are good, but are relatively simple. Here are just a few examples:

- 1) It can select the objects one by one, but cannot select the objects in a region. For example, if users would like to select the cells in a certain region which may include 1000 cells, in Meshlab, users could drag the mouse to select the region, then the objects in this region will be selected. But in MorphoNet, users have to select one by one which needs a lot of time and may not be practical. Also, the inside cells are blocked and hard to be selected.

- 2) I think "neighbor" function is a great idea. But based on my test using the data in Fig 2c, "neighbor" function does not work well. The neighbors for a selected object are not the neighbors in the space. There might be some bugs? Or it picks the neighbors with the nearby labels?

- 3) I believe my computer, browser, and internet speeds are all pretty standard. But as testing the tool in about 30 minutes using their prepared data, it got frozen three times with the warning "not responding". It won't recover after refreshing. I also tried other browser, Google Chrome, it even does not work. After clicking the data, it never showed/loaded the data.

- 4) As I mentioned before, there is no "save" property.

- 5) There are some movies showing examples to visualize different data. But I didn't find the file/tutor to teach how to format and upload the morphological information files.

Developing an online tool is hard. We appreciate the work. There are some local tools can process the data and also have very good packages to visualize the data. We expect the online

tool could match the properties. I believe this tool will be updated better and better. But I feel the current version is still preliminary.

Reviewer #4 (Remarks to the Author):

Overall conclusions

I'd like to preface my general thoughts by saying that I appreciate the difficulty and complexity of the task undertaken by the authors of this manuscript, namely creating software on the web for scientific use cases that also leverages digital 3D data infrastructure. Succeeding in this task requires not just a deep understanding of the needs for scientific applications but also familiarity and expertise concerning modern standards and practices in software engineering, database design, 3D graphics presentation, and data and metadata modeling. And frankly, typically it is scientists who must stretch their skills from the research domain into other areas to accomplish this goal. As a result, creating tools like this is necessary and valuable work.

The authors of this manuscript describe MorphoNet, a product which seems to be described simultaneously in three ways: 1) as a tool for individual biologists to visualize data from multiple phenomic and genetic lines of evidence to derive new insights; 2) as a service for biologists as a community to archive and share multi-faceted phenomic/genetic 3D data on the authors' instance of MorphoNet; and 3) as a server-level software package which is intended to be distributed to and used by other individuals and institutions as a contribution to the field of digital biological data infrastructure tools. The product as described in this manuscript shows a keen appreciation of a novel and important use case for combining 3D phenomic and genetic data, but at the same time the software as described may not meet modern software development and data modeling standards. It is difficult to fully judge this second point due to a significant amount of scientific jargon and a lack of clear technical language, especially since the manuscript language often confusingly mixes the two fields (i.e., using the opaque term "morpho-browser" to describe an in-browser platform for viewing 3D media and related annotations representing biological data). I also have questions over whether current community work regarding data and metadata modeling was consulted during the creation of this platform. For all of these reasons, I think there is a "risk of rapid obsolescence" for this product, as the authors themselves noted in relation to other tools.

To expand on my concerns above, one major omission from this manuscript is whether the MorphoNet software is open source. My suspicion is that it is not, given that the core 3D viewing engine used here, Unity3D, is itself proprietary software. There is certainly a place for proprietary or otherwise closed source software in certain software domains, such as desktop applications, where individual software applications are relatively independent and often do not interact with each other. But MorphoNet is server software, the authors suggest other users could use their software on other server systems. Server software components rely on being able to interact with other software components, and there is a reason why many server software components, even those produced by for-profit corporations, are open source. If MorphoNet is not open source, this severely hampers any potential for long-term use of this product on other systems.

A relating factor to the open source question is that the authors do not indicate where or if the MorphoNet source code is available. Source code for software of this kind should generally be available unless extenuating circumstances exist, and it is my opinion that scientific web software that is proposed to be run by others on external servers should not be published unless the source code is made freely available and easily discoverable on the web. Along with this, there is a lack of detail in the manuscript concerning the technical implementation of this software, and what details are present suggest possible concerns. Software that meets current standards of web development avoids the possibility of obsolescence through being flexible (i.e., able to be displayed through a variety of viewing methods), modular, taking advantage of previously created foundations and frameworks, and using rigorous automated testing. It is difficult to tell whether MorphoNet meets these standards, but if the majority of the application aside from the Unity3D core is custom-written, it is questionable to what degree these standards will be met. These are the benchmarks that determine how difficult it will be for the MorphoNet software to be maintained moving forward

or to be integrated into other environments.

In addition to software development concerns, the limited description given in the manuscript raises the question of whether the authors have consulted and compared with community standards and products when it comes to modeling data organization or creating metadata schema. These are important considerations as they determine whether software systems are capable of “talking” to each other and exchanging data or metadata as required. No mention is made of any other biological digital data web resources (e.g., DataDryad), including those specifically devoted to 3D data (e.g., MorphoSource, MorphoMuseum, Phenome10k). More concerning is that the authors propose a “universal strategy to hierarchically group objects” but do not discuss or seem to account for the significant amount of effort that has already been committed by working groups creating standardized schemas for data and metadata modeling (e.g., PREMIS, PROV, Dublin Core, Darwin Core, this could be a very long list). Is the “universal strategy” suggested here compatible with linked data platforms or the Resource Description Framework?

Creating successful software projects in the current rapidly evolving time period requires integration with previously existing work, as well as building tools in such a manner as to make them more easily integrable with tools to come. A lack of attention to these factors is the underlying reason why much scientific software experiences obsolescence at such a fast rate. As an example of scientific software done correctly, strong consideration for these concepts is why the use of programming languages like R for statistical analyses has been so revolutionary for the practice of science in the last decade. For a concept like MorphoNet to be successful as a digital infrastructure tool, its creators should similarly heed these principles to avoid the same end they themselves observed that many scientific software products meet.

Other General Comments

- Trying out the platform, I seem to be unable to get surface models to rotate in the viewer? Additionally, the layout of the website seems to be not working correctly in Chrome V68.0.3440.106 on OSX Sierra, with the viewer overlapping the header bar with the site logo. Also, on OSX Retina displays, the UI seems very blurry.
- The name MorphoNet seems to be already used:
https://link.springer.com/chapter/10.1007/978-3-642-15754-7_72
- Where is the source code? This is using Unity3D engine, so is this not then open source? Open source software is crucial for collaborative development of scientific software, so a visualization engine wrapped around Unity3D seems like will be very difficult to keep maintained. At the very least, it removes the potential of this software to be incorporated into other projects, which means it cannot be considered a contribution to the evolving field of digital data infrastructure resources.
- There is a fair amount of novel scientific jargon in the manuscript, as well as a general unclear mixing of scientific and technical terms and vocabularies. I am a trained morphologist and a software developer, and so I believe I should be in the audience to understand a manuscript that involves both of these fields. However, I find the language simultaneously too vague and too jargon-heavy (i.e., terms like “morpho-browser” or “web-based morphodynamic browser”). In terms of what I mean by confusion, this manuscript discusses several things: the basic capabilities of the software components the authors use (WebGL being able to load 3D graphics in a browser), what the authors have used these components to achieve (to create an in-browser 3D viewer capable of showing multiple mesh objects and coloring them), and how these tools are meant to be applied (importing data relating to different lines of evidence and visualizing them together for better understanding). But these different levels are not clearly discussed, and the explanations provided often span all of these levels in an unclear fashion.
- Who is the audience of this paper? Is it to advertise the authors’ instance of this platform to scientific users who might use their instance? Or is it to suggest that other bioinformaticians could either set up their own instance or use the components of this software to create new tools? There is not enough technical depth here to address either of the last two audiences, but the authors do seem to want to be writing for them.

Specific Line Comments

- 43: There is no discussion here of the difference between “visualization tools” and biological databases and platforms to share data, and the only previous works discussed seems to be to local

software? No mention of online databases that provide similar functionality (MorphoSource, Phenome10k, MorphoMuseum) or general online databases to share scientific data such as DataDryad, etc. Not even Sketchfab and its annotation capabilities? This does not seem like an adequate characterization of the field as it currently stands.

- 69, 71: This is a very high level description of the implementation of this platform, I would have liked to see much more technical detail. Also again, open source?
- 79: I appreciate and respect the broad scope of the application this software is meant for. That being said, this statement oversells the fact that the authors have implemented a 3D surface mesh in-browser web viewer (of which there are other open-source examples, such as three.js), and can therefore display 3D surface meshes representing many types of biological surfaces. This tool could theoretically also display 3D surface meshes representing buildings or hammers or space shuttles, for instance. More language should be provided to describe exactly how this tool supports a broad range of morphological domains that is not simply encapsulated by the idea that a mesh viewer can display mesh data.
- 100: Did the authors reference any of the many best-practice metadata or data organization standards created by standards working groups? PREMIS, PROV, Linked data models, Resource Description Framework, etc.?
- 101: Does the sentences on this line and following it for the rest of the paragraph essentially boil down the fact that the tool allows display of multiple meshes, with turning visibility on and off per mesh?
- Paragraph 116: In terms often used by the image viewer community, these are annotations, though this word is not used by the authors and no comparison is made to other implementations of annotations. How does this work compare with other current effort to create standardized annotations? Are the annotations created here comparable with other tools?
- 124: Does this mean this tool is capable of reading a variety of data formats or data structures that relate to genomic expression data from RNAseq, etc.? If so, what are these formats or standards? Are these all data standards particular to this project?
- 128: What databases are connected via APIs? What are the “universally formatted files”?
- 283: This sentence describes a fundamental model for all web resources, I think the explanation should be more elaborate to connect this statement with the specific tools discussed.
- 286: Asking users to specifically use one browser does not reflect modern web development standards
- 287: This is an unusual request for users (and is also not explained how this is done), given that other websites displaying 500MB meshes such as Sketchfab and MorphoSource have no similar requirement
- 291: Is this section a description of the implementation of the website or a guide for how to import data into the tool? The narrative of these paragraphs is confusing
- 300: And yet the authors recommend only using Firefox?
- 310 (whole paragraph): A general issue here is that this website (without logging in) does not seem to provide any evidence for the technical claims regarding any of its functionality, including the API. There is also no publicly accessible documentation. Also, can the software have a GNU license when using Unity3D as a core component?
- 391: Most common mesh format is .obj? It is certainly a common file format, but I imagine something like .stl is more common.

Reviewer #5 (Remarks to the Author):

I am very excited to start using MorphoNet! I have helped to create some browser-based biomedical image viewing interfaces, and am used to using the ascii VTK format and visualizing with ParaView, Meshlab, and other interfaces that do not permit easy online viewing, combining, and sharing. The authors deserve particular credit for making MorphoNet open source, online, and free, with open, widely used standards for its format and API. I was excited to watch all of the videos to get a practical sense not just of MorphoNet's features, but of the practical manner in which a user interacts with the front-end.

I look forward to converting my own (.nii and .vtk) brain imaging data and brain image analysis software (mindboggle.info) output to .obj to enable people to explore with MorphoNet. I would also be interested to have MorphoNet host these public data for use by the scientific community, and

hope many others with large datasets will do the same.

Three suggestions:

(1) Please provide guidance for a reader/user to reformat (e.g., VTK) surface mesh data (you mention Meshlab) or to create .obj meshes from (e.g., dicom, nifti, stl) volumetric data for use in MorphoNet.

(2) lines 217-218: "The dataset position and orientation in space can be reset to default values, and the default orientation can be set (only by dataset owner)."

It would be very nice to capture a view of a dataset (orientation, crop, zoom, and any modifications to colors, transparency, etc.) and share that view as a starting point for another user (more than just as a screenshot).

(3) I appreciate the dynamic changes in color-coding and transparency of objects. Would it be possible to view the meshes or surface normals, or to cull back surfaces of objects? It would also be nice to have a static view with 3-D quiver plot to contextualize movement/change between frames.^[1]_{SEP}

Edits:

- * 17 biology, at scales ranging from the molecule to the functional organ. To support this big-data
- * 18 revolution, we have developed a concept of generic web-based morphodynamic browser to
- * 26 to the development -> with the development
- * 27 specimen -> specimens
- * 34 need of -> need for
- * 71 clouded -> cloud-[based, stored, accessed,...]
- * 90 chambers volume -> chamber's volume
- * 91 Human -> human
- * 99 universally-formatted files, that -> universally formatted files that
- * 171, 173, 176 featuring -> features
- * 178 discussions on -> discussions of
- * 213 numerical id -> numerical ID
- * 224, 226, 231, 234, 363, 405, 444, 451, 463, 470, 479 associated to -> associated with
- * 224 Information are -> Information is
- * 225 are color labelling applied -> are color [labels applied, label applications]
- * 240 downloaded in as -> downloaded as
- * 243 Specific objects interactions -> Specific object interactions
- * 245 cluster group -> cluster a group
- * 253 expression on data is -> expression data are
- * 269 as .mp4 -> as an .mp4
- * 283 composed by -> composed of
- * 286 on any Internet browsers -> on any Internet browser
- * 293 dataset mesh -> a dataset mesh
- * 300 codes -> code
- * 301 browsers -> browser
- * 303 on application/request -> by [request, application/request]
- * 306 etc.). -> etc.).
- * 314 users -> user
- * 317 data is uploaded on -> data are uploaded to
- * 318 each new meshes in -> each new mesh to
- * Table 1. MorphoNet Users Rights. -> MorphoNet User Rights.
- * 345 to MorphoNet -> to the MorphoNet
- * 357 nor -> or
- * 372. a public -> public
- * 379 advantage from -> advantage of
- * 381 his -> its
- * 385 the id -> the ID

- * 390, 463 under -> in
- * 465 overimposed to -> superimposed on

Supplementary:

- * color coded -> color-coded
- * names, corresponding to each organ, is -> names, corresponding to each organ, are
- * along to different -> along different
- * three dimensional -> three-dimensional
- * VTK library -> the VTK library

[L]
SEP

Cheers,
Arno Klein

List of main modifications and additions

1. Asked by all reviewers

1.1. The open-source distribution of MorphoNet code and its licensing.

We distribute the whole MorphoNet code open-source on a dedicated MorphoNet GitLab (<https://gitlab.inria.fr/efaure/MorphoNet>) . In this GitLab users may find the complete code, the documentation required to install a MorphoNet instance on any server, the license (CeCILL, a GNU GPL license) for the code release.

1.2. Connection with other software and data

In the MorphoNet GitLab (<https://gitlab.inria.fr/efaure/MorphoNet>) users may also find several technical tools to illustrate and simplify the use of MorphoNet and its connection with other tools, softwares and formats. These tools consist of:

- A. Three APIs (Python API, ImageJ API and ANISEED API) to allow a direct connection of MorphoNet server with, respectively, Python scripts, ImageJ/Fiji software and the genetic database ANISEED. The Python API is also accessible at <http://www.morphonet.crbm.cnrs.fr/HELP/HelpAPI.php> after login;
- B. Several python scripts to automatically convert multiple 3D image formats (including the .nii format requested by referee 5), .vtk and .stl mesh formats and .csv information format to the dedicated MorphoNet data and metadata formats. These converters are also accessible at <http://www.morphonet.crbm.cnrs.fr/Converters/Converters.php> after login;

Through the ImageJ/Fiji API we have created an ImageJ/Fiji MorphoNet plugin, which allows with simple mouse clicks the upload of images visualised in ImageJ and properties calculated by built-in ImageJ libraries directly to the MorphoNet database.

1.3. Specific technical and scientific examples of MorphoNet use

In the MorphoNet GitLab (<https://gitlab.inria.fr/efaure/MorphoNet>) users can also access three detailed tutorials illustrating the use of MorphoNet to address specific scientific and technical problematics. These tutorials are also accessible at http://www.morphonet.crbm.cnrs.fr/Tutorials/MorphoNet_Tutorials.php after login:

- Tutorial 1 shows, starting from 3D segmentations of developing embryos, how to produce surface meshes in the MorphoNet format, how to upload them to MorphoNet server by creating a new dataset and how to employ MorphoNet, Python and the dedicated API to calculate, visualise and explore quantitative dynamic properties of cells and tissues.
- Tutorial 2 illustrates step-by-step the use of the dedicated MorphoNet plugin in ImageJ which permits by simple mouse clicks a direct upload to the MorphoNet server of surface meshes and qualitative and quantitative information calculated by built-in libraries of ImageJ.
- Tutorial 3 exemplifies how MorphoNet and its API can be of great help during segmentation routines in identifying and correcting some of the most common issues of volumetric reconstructions: over-segmentations, under-segmentations and missed temporal links between objects of a timestack.

1.4. Modifications in the main text. In particular with a paragraph commenting on the use of MorphoNet for specific research applications

We have modified the main text, the Methods section and the Supplementary Material following the referees' suggestions. In the following we provide the complete list of these modifications. All modifications can be tracked as they are written in red font in the main text, Methods and Supplementary Information. All modifications in the text to comply with reviewers' requests are stated explicitly in our detailed answer to each referee.

Modifications in the Main Text:

- A. (L173) Inspired by the aforementioned tutorials and to comply with reviewers' requests, a whole new paragraph briefly illustrating the use of MorphoNet for specific

research and technical problematics has been added before the conclusions in the main text.

- B. (L53) We changed a sentence to comply with reviewers' comments.
- C. (L58) Two new sentences have been added.
- D. (L84) A sentence has been added.
- E. (L111) A sentence has been modified to improve its clarity.
- F. (L115) A sentence has been added.
- G. (L140) Two sentences have been modified to integrate further information.
- H. (L153) A sentence has been added.
- I. (L157) A whole new paragraph has been added to clarify the built-in calculations of MorphoNet and its connection to external platforms.
- J. (L201) A sentence has been added about the perspective of MorphoNet development
- K. All typos detected by referee 5 have been amended.

Modifications in the Methods:

- A. (L339) In the Implementation section a sentence has been added to better describe the structure of MorphoNet.
- B. (L347) In the Implementation section two sentences have been modified to clarify a detailed assessment on the limit in terms of dataset size.
- C. (L448) In the Morphological Datasets (Mesh) section, a sentence has been modified.
- D. (L459) In the Morphological Datasets (Mesh) section, a whole new paragraph has been added to discuss memory limitations of the WebGL.
- E. (L554) Code availability was added to follow Nature Research requirements.

Modifications in the Supplementary Material:

- A. All typos detected by referee 5 have been amended.

2. Technical modifications in response to comments of one or more specific reviewers

2.1. Possibility to represent quiver lines on datasets (Reviewer 5)

We have replied to the referee by explaining in details how to use MorphoNet features to represent quiver lines on a dataset. We also provide now an example of quiver lines visualisation on the dataset "**Phallusia mammillata embryo (Wild type, live SPIM imaging, stages 8-17)**".

2.2. Representation of mutations and/or perturbations (Reviewer 2)

In addition to our detailed answer to his/her concern, we have now also connected MorphoNet with the genetic mutants database of ANISEED, to show how one can project information about altered genetic expression onto ascidian embryonic morphologies.

2.3. Evaluation of the maximal number of objects for visualisation and interactions in MorphoNet (Reviewer 1)

We have performed multiple tests of upload, visualisation and interaction of several simulated datasets. Each dataset had a specific numbers of elementary objects distributed over different timepoints. We could thus address the specific limits imposed by the WebGL technology as of today: for an optimal visualisation and interaction experience, each dataset timepoint should not contain more than 20000 objects. Note also that, in case of massive datasets, this limit can be respected by defining larger elementary objects (e.g., tissues instead of single cells). We have replied to the reviewer and added a sentence about this limit in the Methods section.

2.4. Addition of new public datasets to the MorphoNet server (Reviewers 1 and 4)

We added 9 new public datasets following reviewers requests and comments:

- A. **Danio rerio embryo (Wild type, live confocal imaging)** and **Danio rerio embryo (Wild type, live light-sheet imaging)**: Zebrafish embryonic development imaged by confocal and light-sheet microscopies. Data published in J. Stegmaier et al., *Real-Time Three-Dimensional Cell Segmentation in Large-Scale Microscopy Data of Developing Embryos*, *Developmental Cell* 36, 225–240 (2016)

- B. ***Drosophila melanogaster* embryo (Wild type, live confocal imaging) and *Drosophila melanogaster* embryo (Wild type, live light-sheet imaging):** *Drosophila* embryonic development imaged by confocal and light-sheet microscopies. Data published in J. Stegmaier et al., *Real-Time Three-Dimensional Cell Segmentation in Large-Scale Microscopy Data of Developing Embryos*, *Developmental Cell* 36, 225–240 (2016)
- C. ***Mouse* embryo (Wild type, live confocal imaging) and *Mouse* embryo (Wild type, live light-sheet imaging):** *Mouse* embryonic development imaged by confocal and light-sheet microscopies. Data published in J. Stegmaier et al., *Real-Time Three-Dimensional Cell Segmentation in Large-Scale Microscopy Data of Developing Embryos*, *Developmental Cell* 36, 225–240 (2016)
- D. **Voxel Man (Frozen body of adult male, photographic cross-sectional images, CT and MRI):** Virtual reconstruction of the torso and inner organs of a human body, based on a stack of 1878 photographic cross-sectional images of a frozen male body, along with corresponding CT and MRI images. Published in A. Pommert et al., *Medical Image Analysis* 5, 3 (2001), 221-228.
- E. ***Canariomys bravoi* (Fossil, computerized microtomography):** three-dimensional reconstruction of the skeleton of the giant rat of Tenerife (Canary Islands, Spain) *Canariomys bravoi*. Data taken from the online database MorphoMuseum.
- F. ***Macropus eugenii* (Tamar wallaby) pouch young (Wild type, scan):** Scan of a *Macropus eugenii* (Tamar wallaby) pouch young. Data taken from online database Phemome10k.

3. GitLab account

The access to this GitLab account is at the moment restricted to authenticated users only. It will be made open-access upon acceptance of the manuscript.

Address : <https://gitlab.inria.fr/efaure/MorphoNet> (click on **Standard** authentication)

Login : morphonet

Password : M0rph0\$e!

At the moment, access to MorphoNet requires users to register online. Once the work is published, public datasets will be accessible by any unregistered user. To facilitate the assessment of the tool, we have created a special referee account on MorphoNet, which gives reviewers the necessary rights to access a large part of the information stored on the server and to exploit the full functionality of MorphoNet:

url: <http://www.morphonet.crbm.cnrs.fr/>

login: reviewer.access

password: review

Detailed answers to Reviewers

Reviewer #1 (Remarks to the Author):

R1.1 It's a great idea to have an online browser of 3D and 4D image data to allow researchers to share and interrogate data in an interactive way. This work attempts to provide this for segmented images. I think the authors should be commended for attacking this issue. The manuscript is clearly written and makes a strong argument for why such a facility is required.

We thank the referee for his/her kind words and for the high consideration of our work.

R1.2 I found using MorphoNet reasonably straightforward. The best data available to reviewers to try this out is currently the Phallusia embryo and it was relatively easy to get an idea of the basics of the morphogenesis of this simple embryo. Using the browser would be improved if combinations of the main dropdown menus were simultaneously available – for example I want to interact with the plane crops available under the Dataset dropdown while simultaneously viewing the list of genes whose expression profiles I had selected in the Genetic dropdown.

That is a very interesting idea which will surely make the MorphoNet experience better. We have a series of technical improvements of this kind in mind, which will be implemented for the next software release.

To encourage such suggestions of improvement, we have implemented the possibility for users to provide feedbacks and suggestions directly from the online interface.

R1.3 It would be good to see how successful this tool is when the embryo or organ is more complex. The zebrafish embryo example also stops at a relatively simple stage of development and the Drosophila embryo isn't a complete embryo so it impossible to gauge how this will deal with more sophisticated data. Can the authors say what the maximum number of cells and timepoints that could be handled by MorphoNet is? They do give some suggested maxima for file size and elementary objects but these are a bit abstract to me (someone that doesn't use segmented data), how does that translate to cells and timepoints?

We have tested the upload and interaction with several simulated datasets, ranging from a few to several thousands objects distributed over several timepoints, roughly following the cell number of zebrafish development. We have thus addressed the WebGL memory limits in terms of cell number: for an optimal use, the total number of objects of a dataset should not be higher than half a million, considering not more 20 thousands cells per time step and not more than a few hundred thousand time steps. These limits are now quantitatively stated in the manuscript where we have added in the Morphological Datasets section of the Methods the sentence (L459): *“For an optimal experience when interacting with the dataset, the total number of objects should not be higher than half a million. This is due to the intrinsic limits to the WebGL memory as of today.”*

R1.4 I also tried out the Human brain anatomy and abdomen datasets as I teach these things to undergrads. I can't say I found these very useful. There are other tools and datasets available that are better (as least from the teaching point of view).

The idea of MorphoNet is indeed to allow any user to upload his/her data to shared databases. We have provided some datasets as examples, and we are sure far better data have been produced and used. However our goal is not to build a complete database ourselves, but rather to give the possibility to large communities to share their data and analyses in a straightforward way. Although we have not explored the enormous amount of data available online, we are very happy that the referee liked the software enough to consider it for teaching. We regret that the dataset provided was not adequate. In response, we now provide a second Human body datasets from Voxel Man (<https://www.voxel-man.com/segmented-inner-organs-of-the-visible-human/>) for visualisation and interaction. But, analogous to the publication of a genome browser, the data provided here are mere examples to illustrate the power of our tool.

We have also explored several online tools for visualizing and teaching human anatomy (such as <https://human.biodigital.com/index.html>) or specifically dedicated to the brain (such as <http://www.g2conline.org>), and we agree with the referee on the fact that these tools, specifically developed to interactively show the human anatomy, are naturally more adequate for teaching such discipline, thanks also to the well-curated explanations on each anatomical element. We stress however that MorphoNet is fundamentally different in its spirit, since it offers the possibility for a teacher to upload his/her own dataset and even his/her own experimental results.

R1.5 Of course the information available in these segmented images will only ever be as accurate as the segmentation of raw data and the annotation made by the uploading lab. So it's not really possible to know how much confidence to put on the information that you are trying to interrogate. Without the raw data or at least some estimate of the limitations/accuracy of the segmentation this will be a worry.

The reviewer raises an interesting and fundamental question in the field of biological image analysis.

MorphoNet has been developed as a visualization tool used downstream of the segmentation step. It is in this sense analogous to a genome browser displaying gene models that have been generated externally. That said, we have over the past few months discovered that MorphoNet can be of great help to evaluate the overall quality of 3D+t segmented data because it allows to project properties onto the segmented data to evaluate their morphology. For instance cell volumes or cell lineages during embryo development can be easily visualised, pointing to abnormal cell volumes, interrupted lineage branches, or premature cell divisions. This led us to extend the functionalities of MorphoNet to correct segmentation issues, thus improving the overall quality of any segmentation strategies.

To show this, we now provide a detailed tutorial (tutorial 3 at http://www.morphonet.crbm.cnrs.fr/Tutorials/MorphoNet_Tutorials.php, see Section 1.3 of the **List of main modifications and additions** at the beginning of this letter) showing how MorphoNet and its API can help identifying and easily correcting segmentation and tracking errors.

R1.6 Will MorphoNet be a success? There's probably only one way to find out and that's to make it widely available and then monitor how many labs upload their data and how many other people successfully interrogate the shared data. I think it's definitely worth finding this out for MorphoNet.

We thank the referee for his/her opinion on MorphoNet. There will actually be two important measures of the success of MorphoNet: how many communities set up their own server, and how many use the first server we set up. The code will be open source and distributed on a GitLab (<https://gitlab.inria.fr/efaure/MorphoNet>). This will allow us to monitor the number of downloads of the application and assess how many communities have decided to set up their own server. In addition, we can also monitor the user connection statistics on the MorphoNet server we implemented.

Reviewer #2 (Remarks to the Author):

In this paper Leggio et al. present a very innovative online computational tool -called MorphoNet- to explore complex dynamic morphological datasets using user-friendly interfaces with intuitive representation of biological entities, across organizational scales.

This project was inspired by the emergence of large scale datasets from state-of-the-art imaging technologies and the analogy with the concept of genome browsers, which offer a unifying platform to navigate other complex, sequence-based, datasets.

In this regard, the creation of MorphoNet is timely, and its potential impact could hardly be overstated.

R2.1 Among the specific merits of the MorphoNet platform and the manuscript, the authors provide very strong justifications for a web-based analog to genome browser for morphodynamic data; they used clever formatting of segmented image datasets to allow portability; the tool allows for biologically relevant hierarchical groupings, and for computational integration of other datasets (such as sequencing data) onto a morphodynamic reference; finally, MorphoNet provides powerful visualization tools and even analytic functionalities.

We thank the referee for his/her very positive assessment of our work.

I certainly commend the authors for this ambitious and important work, but nevertheless wish to provide some feedbacks, which I aim to be constructive, especially regarding the current format of the paper.

GENERAL CONCERNS

R2.2 First and foremost, the paper is well written, but from a very general perspective, which makes the specific aspects of MorphoNet sometimes hard to grasp. It almost reads more like an "infomercial" than an actual scientific article, and it is not entirely clear how one would use MorphoNet to address specific problems. Arguably, the tool's potential is too vast to illustrate comprehensively, but perhaps using MorphoNet to address a particular set of biological problems, while navigating and introducing the various features of the software would better demonstrate the power of the tool.

Along the same line, the methods are written as a tutorial, but a verbose one, without the specific indications on how to perform indicated tasks. Best would be to have a proper methods section describing the main methods used to build MorphoNet, and then write a detailed tutorial to publish as supplemental information for prospective users. Here too, illustrating with specific scenarios would probably be more helpful than general statements such as “User can also [do this or do that]” without clear indication about how to.

As the reviewer points out, we did our best to design MorphoNet so that it is as generic as possible and tried to convey this genericity by adopting a general perspective when writing the manuscript. Instead of focusing on certain specific uses, with the risk of giving a partial and incomplete view on the potential of this platform, we have chosen to present a detailed overview of its features. The idea to create examples of use of MorphoNet to address specific biological problems is ingenious and we thank the referee for that. We have followed his/her suggestion and provide now three Tutorials (see Section 1.3 of the **List of main modifications and additions** at the beginning of this letter) available online at http://www.morphonet.crbm.cnrs.fr/Tutorials/MorphoNet_Tutorials.php after login.

SPECIFIC CONCERNS

R2.3 It is not clear in the main text how the data used to create the reference models were collected, and processed. Although this information is available deep into the “METHODS” section, this could be addressed by providing very specific examples of using MorphoNet to address biological problems of interest.

MorphoNet has not been conceived as a image-processing platform, but rather as a visualisation, interaction and sharing tool for segmented images and quantitative analyses on them. As such, it is not the scope of MorphoNet nor of our manuscript to provide details on how raw images are processed to achieve segmentations: a vast literature is available on this subject, consisting of several methods, each specific for a particular scientific domain. In the same spirit, one would not expect a paper developing the concept of genome browser to explain how gene models were created. What matters most here is that the format of the data that can be uploaded in MorphoNet is standardized and clearly described, which we hope we managed to do.

About the use of MorphoNet to address specific biological questions, tutorial 1 (see Section 1.3 of the **List of main modifications and additions** at the beginning of this letter) shows how MorphoNet can be efficiently employed to explore the dynamics of cell shapes during gastrulation in ascidians.

R2.4 Figure 3 shows gene expression patterns -presumably from in situ hybridization data- mapped onto embryo models, and the text mentions RNA-seq data, which is inherently more quantitative. I was wondering whether the latter can be represented as heatmaps to convey the expression level as well as the pattern, only to find answers deep in the “METHODS” section.

Indeed, heat maps can be used to visualise quantitative expression levels. This is an important feature, which we have now decided to mention in the main text (L139): *“MorphoNet offers the possibility to upload gene expression data from in situ hybridizations, cell- or tissue- specific RNAseq, Starr-seq, or chromatin signatures and to superimpose them onto a 2D or 3D scaffold as either boolean (color selections) or quantitative (heatmaps) information.”*

Although we do not yet have quantitative single-cell expression patterns for the datasets we collected and cannot provide a figure exemplifying this feature, the use of heatmaps to represent the distribution of another kind of quantitative geometric properties is exemplified in the aforementioned tutorial 1.

R2.5 A more conceptual concern refers to the handling of perturbation datasets, considering that experimental perturbations may alter the morphodynamics used as reference. Here too, a test-case would better illustrate how MorphoNet can be used to integrate both the molecular and morphodynamic effects of perturbations than the general statements in the text.

In order to explore such perturbations in MorphoNet, one has two possibilities: one can simply create a new dataset, with the associated perturbed morphology, and treat it as any other dataset by projecting morphological information onto it; or one can calculate (in Python, in ImageJ, manually or in any other plausible ways) the desired information on the perturbed morphology and project it onto the wild-type. This shows where the wild-type properties have changed following the perturbation. Let us briefly consider a specific example: a perturbation which alters the volume of certain cells of one embryo. Under the hypothesis that one is able to identify the same

cells in the perturbed and unperturbed embryos, through the python API one can calculate the perturbed volumes, download the unperturbed volumes and upload a new information, for example the difference or the ratio of the two. In such a way, a heat map can easily show which cells are most affected by the perturbation. Note that, if cell-to-cell identification is impossible between perturbed and wild-type embryo, one could do the same at the level of tissues (or any other selected groups of cells whose identification between mutant and wild-type is possible). In this way, the combination of MorphoNet interface and its Python API permits the exploration of qualitative and quantitative effects of perturbations at many different scales.

R2.6 Along the same line, it is not clear at all how MorphoNet can be used to compare morphodynamic processes across species. What would the morphodynamic reference be for different species? Here too, specific examples would be helpful.

The answer to the question of the referee depends on the species to compare. For instance, between two ascidian species the common reference across species would be the cell lineage and therefore the cell names, at least up to the gastrula stage. Again, thanks to this reference one could project cell properties of one species onto the cellular scaffold of the other, or represent as a heatmap the variation of a property (say, the time between consecutive divisions) of corresponding cells in the two species to investigate whether interspecies variability is localized on certain tissues and/or certain developmental stages or is roughly the same everywhere in the embryos. As above, if cell-to-cell identification is not possible between two different species one could change scale and compare tissue or organ properties.

To point this out in the main text, we have added a sentence shortly (L152): *“Interspecies variability of relative tissue volume of developing embryos can for instance be pre-calculated and visualized as a heat map, to show whether it is localized on certain tissues and/or certain developmental stages.”*

More generally, a comparative study on variability/stereotypy of the embryogenesis of several different species has, to our knowledge, not yet been performed extensively at single-cell level. One of the main obstacles to this study is the fact that labs and communities are often specialized on one or few species and produce images and quantitative data with specific formats, difficult to share and be used by other research networks. MorphoNet, however, represents the linking environment within which all these results can come together under a unifying format, a first fundamental step to trigger comparative studies. These analyses would in addition require powerful tools to visualise datasets and project variability measures onto them. MorphoNet provides the adequate tools to address these needs and will also be extended in the future to address more aspects related to individual-to-individual comparison or developmental variability within and between populations.

R2.7 The last part before the summary (LL 133-137) is important but explained very succinctly. Are the calculations for data analyses performed within MorphoNet? Or is it interfaced with popular data analysis software like Python or R? It might be useful to explain how MorphoNet can be integrated with these other popular softwares (e.g. ImageJ/Fiji “upstream” of MorphoNet; Python and R “downstream” of MorphoNet).

The referee is right, we did not specify with sufficient clarity how these calculations can be performed. There are a few simple calculations (average, standard deviations, normalisations of quantitative information uploaded to the MorphoNet server) which can be performed by the MorphoNet server and exist as built-in options in the MorphoNet interface. Due to the web-based structure of the tool, more costly calculations on the server would affect the dynamic interaction of the user with the dataset.

For more complex calculation, as detailed in Section 1.2 of the **List of main modifications and additions**, we provide interfaces harnessing external tools to the computation of properties: we have created a Python API (<http://www.morphonet.crbm.cnrs.fr/HELP/HelpAPI.php>) and ImageJ plugin (also discussed in tutorial 2 of Section 1.3 of the **List of main modifications and additions**) which can directly compute and upload the calculated property to the MorphoNet server. The user also has the additional option to use any other external tool provided the end result is formatted according to our precise information format (see <http://www.morphonet.crbm.cnrs.fr/HELP/HelpFormat.php> for guidelines). For this, the user can use our converters (provided here: <http://www.morphonet.crbm.cnrs.fr/Converters/Converters.php>) to create .txt files in the specific MorphoNet format from pre-calculated properties and upload them directly from the online interface. We added a sentence in the main text to explain this (L157): *“A few simple calculations (average, standard deviations, normalisations) exist as built-in options in the MorphoNet interface. Due to the web-based structure of the tool, more costly calculations on the server would affect the user dynamic interaction with the dataset.”*

However more complex calculations can be performed externally. The Morphonet Python API and ImageJ plugin provide an easy interface to upload any Python-based or ImageJ-performed calculations directly to the MorphoNet server. Otherwise the user has always the possibility to create .txt files and upload them from the online interface."

R2.8 Similarly, it is not entirely clear from the paper how large genomics datasets would be integrated with morphodynamic data using MorphoNet.

Cells at a given time point can be ascribed all sorts of properties, including genomic properties. For instance, RNAseq experiments for each gene, ATAC-seq status for genomic loci, etc. MorphoNet, thanks to its dedicated genetic format (<http://www.morphonet.crbm.cnrs.fr/HELP/HelpFormat.php>) can integrate this kind of information and visualize it onto the cell structure of each dataset.

We have currently integrated two genomic databases in MorphoNet. At the small scale, a manually-curated gene-expression database for Arabidopsis meristem (private data, not available to referees). At the large scale, we have integrated MorphoNet with the extensive ascidian genomic database ANISEED into. This integration is done via the specific Aniseed API and through a Python code, which could transform any genomic database into the MorphoNet format. To address the referee's remark, we provide now the Python code used for the Aniseed transformation. We have not yet tested the integration of other genetic databases but we expect that, after the publication of MorphoNet, we will get in touch with other research communities and other types of databases. Depending on the structure of new genetic data, the development of new integration strategies with the MorphoNet database might be needed: this is also one of the reasons for us to provide the MorphoNet code open-source. The API for ANISEED integration is available on the MorphoNet GitLab at <https://gitlab.inria.fr/efaure/MorphoNet>.

MINOR CONCERNS

R2.9 Abstract: suggest "research and education" instead of "research and teaching"
We have followed the referee's suggestion and changed the text accordingly.

Reviewer #3 (Remarks to the Author):

In the manuscript "MorphoNet: An interactive online morphological browser to explore complex multi-scale data", Leggio et al introduce a web-based tool to interactively visualize complex image datasets in a form of surface mesh with additional morphological information file as the input. They would like to develop a generic tool matching the generality and intuitiveness of genomic browsers.

R3.1 Overall, online tool is a great approach especially with sharing property. Developing such online tool may meet many challenges. I think with the prepared data and external information matching the data, this is a very useful tool for teaching and presenting the data.

We thank the referee for recognizing the potential of MorphoNet for teaching and dissemination.

R3.2 But in the manuscript, there is no strong evidence to show how this tool has biological meaning in research. For example, there is no example to show any new biological findings by using this tool.

In this paper, we chose to present a tool that is currently lacking in the community for easy manipulation and sharing of 3D(+) organismal complex data. We initially designed this tool and put a lot of effort in its development to answer critical needs that appeared in our own research. However, an important number of enthusiastic feedbacks from both animal and plant biologist colleagues, has encouraged us to push Morphonet development to professional standards and to publish it.

In addition, first major biological results that can be obtained using such a tool are about to be published in a series of companion works. We recently submitted a work on Ascidian development that makes critical use of the Morphonet browser. The work is available on BioArchive (Guignard, L. et al. Contact-dependent cell communications drive morphological invariance during ascidian embryogenesis, *bioRxiv* 238741 (2017)).

Two additional works on the mechanics of oriented cell divisions and on an atlas of gene expression in Arabidopsis meristem are being currently written and have made largely use of MorphoNet for both analyses and discoveries.

As a matter of fact, this publication is aimed at disseminating Morphonet as a new tool to ease the process of discovery on such complex biological data.

To better illustrate the biological relevance of using Morphonet in such a work, we now discuss more thoroughly in this Morphonet paper how the dynamics of cellular shape changes during ascidian gastrulation and how this can be evidenced using Morphonet. The complete and detailed use of MorphoNet in addressing this question is shown in Tutorial 1 (see Section 1.3 of the **List of main modifications and additions** at the beginning of this letter). We show that MorphoNet can efficiently be employed to explore the complex 3D dynamic shape changes occurring during ascidian gastrulation: by the visual online interface, cells of interest can easily be identified and selected for quantitative analysis. Through the dedicated Python API, such a selection can immediately be read by a python script, which calculates specific quantitative characterisation of cellular shape changes (external surface area, cellular elongation). Such a quantitative analysis can then, again via API, be uploaded to the MorphoNet server. The visual interface can then again be used to visualise, in the form of a heat map projected on individual cells, the change in time of these properties. It is then possible to visually identify the onset of gastrulation as the beginning of the first shape-change process (apical constriction of endodermal cells), and straightforward to note how the second shape-change process (apico-basal shortening) begins after the first one is complete. MorphoNet thus allows easily to explore a complex 3D+t morphodynamical process which has been an open question in the community of ascidian developmental biology for decades.

R3.3 There do exist some tools not only can visualize data but also can process and measure the data (e.g. drishti). Meanwhile, with many constrains with the input files (e.g. surface mesh only, file size limitation, limited number of objects, morphological information has to be extracted by other imaging tools) and relatively simple properties, I think the current version is still preliminary and not generalized enough to satisfy broader users. I believe users do expect the online tool could match the local tool in terms of functions and properties.

We absolutely agree that several excellent tools are available for local morphological analyses and visualizations. Many of these tools are used on a daily basis, even by us.

MorphoNet is complementary with such tools. Notably, applications that run locally on users' computers, by definition, are made to interactively visualise datasets locally. They cannot be used to share data and related quantitative information with users worldwide, which is one of the major reasons why we developed MorphoNet. MorphoNet, which does not require any installation, allows to share complex morphological datasets, related calculated information and research results within and between research communities. To the best of our knowledge this is currently not possible with any other existing tools.

Sharing these very large datasets through the web obviously imposes technical constraints, because of memory limits and bandpass. This limitation is however not intrinsic to MorphoNet but rather due to the characteristics of built-in web browsers. The web-based and share-oriented vision also imposes more constraints on the format employed for morphological datasets and tracks than for dedicated local tools.

We chose the surface mesh as format for morphological data because contrary to images.

Finally, we would like to stress that we have designed simple property types that cover almost entirely any kind of possible morphological or genetic tracks that we have met or conceived.

More specifically:

- Segmented images (2D, 3D with or without time information) can easily be transformed into surface mesh. For this task, the revised manuscript provides Python scripts (<http://www.morphonet.crbm.cnrs.fr/Converters/Converters.php>) and an ImageJ plug-in (the API for which is provided at the MorphoNet GitLab (<https://gitlab.inria.fr/efaure/MorphoNet>); see also the tutorial 2 at http://www.morphonet.crbm.cnrs.fr/Tutorials/MorphoNet_Tutorials.php to perform this conversion automatically).
- Based on our tests, the number of objects is limited to roughly 25 thousands per timepoint. Thanks to the flexibility in the definition of elementary objects, however, this number does not represent a fundamental limitation.

Finally, we expect this first release of MorphoNet to rapidly develop thanks to the distribution of the MorphoNet open-source code at the GitLab address <https://gitlab.inria.fr/efaure/MorphoNet>.

Below, comments are more details.

R3.4 I understand that using only surface mesh is to save memory and to speed up. Only visualizing the surface mesh of a single object is not novel enough as too many tools can do it. To take advantage of this tool, the input data requires the volume segmentation first. For example, with the CT scans, the segmentation has to be done by other software with the volume data as input. After segmentation, different parts/objects will be labeled differently in a volumetric format. Many software such as drishti can visualize it in a similar way, and also can render and measure this volumetric data. But to be able to use MorphoNet, the users have to extract the surface mesh for each part from the volumetric data, assign labels, and then format it in a certain way for the input file of MorphoNet. In other work, it needs a lot of additional data processing to prepare the input files. If other tools can visualize it similarly without further data processing, the users may need stronger reasons to use MorphoNet.

We agree with the referee that other local tools, such as drishti, may offer more flexibility. However, the aims of tools deployed locally and web-oriented ones are not the same: as commented in details previously, the spirit and the interest of MorphoNet lies in its intuitive web-based structure representing a sharing and linking environment for different communities.

Using standardized formats is also a defining feature of genome browsers (think of Fasta, GFF3, bigwig, etc) and we reasoned along the same lines. But we took good note of the referee's point that conversion to mesh and to the right format should be made as simple as possible. That is why we have developed Python scripts and an ImageJ plug-in to generate meshes automatically from images. We have also developed converters to import data from other formats such as csv, vtk and stl directly into MorphoNet (<http://www.morphonet.crbm.cnrs.fr/Converters/Converters.php>).

In the manuscript, the authors point out there are three major limitations for other existing morphological visualization platforms. However, I only partially agree with this current version of MorphoNet.

R3.5 "...not designed to integrate the user's data with external sources" (line 48). I agree that the highlight of MorphoNet is that it can show the group information with those external information as input. It can hide and show any selected objects. The main concern is the external morphological information needs to be extracted by other tools and also needs to be formatted in a required way. This might limit the generalization. Many tools that can extract the morphological information also can visualize the data in the similar way.

Similarly to genome browsers, the main scope of MorphoNet is to project morphological information on datasets and, above all, to allow the sharing of information and datasets within and between communities, thanks to its online interface and its unifying morphological formats.

As commented above, we have now implemented the possibility to automatically convert other standard formats into the specific one of MorphoNet (<http://www.morphonet.crbm.cnrs.fr/Converters/Converters.php>) and an ImageJ plugin which automatically connects ImageJ with the MorphoNet database.

Unlike the reviewer, we think that this strategy increases the generality of the tool, as it will allow people from very different communities (from paleontologists, to ecologists and embryologists) to upload very different types of images, converted to the same format.

Finally, we do not know of tools installed locally that can be integrated with genetic databases.

R3.6 "...local analysis tools can hardly be used to share research data" (line 50). Being able to share the data is a great idea, but I realize after interactively visualizing the data, the current version cannot save it. For example, what if the user would like to share the file after selecting and coloring. The current version cannot do it. In other words, the file is just shared as the file. Thus the tool is functioning as a drive or cloud similar to Google Drive and Dropbox. I think it is good to be able to save the script of what the user did and share it so that other users in the group could run and see it.

Dropbox and Google Drive allow to share individual files and nothing more. This is obviously very different from MorphoNet, which allows to visualize and interact with whole structures of data and metadata. The sharing option does not simply share metadata in one isolated file, but rather shares the complete connection of such metadata to its corresponding dataset, and specifically to each object in the dataset.

Yet the referee is right that MorphoNet users cannot currently save a specific view on the dataset and return to this view upon login. This feature will be implemented in the next platform release.

Whole processes of user interaction with datasets, videos and snapshots can however be recorded and shared through videos made directly from the online interface.

Of note, most genomic browsers do not allow to return to a specific view (locus, magnification, etc...), upon login. However, they allow saving sessions with a personalized configuration of tracks that are shown by default upon login. Morphonet also has a sessioning system although it is currently less sophisticated than genomic browsers:

- 1) users can upload private tracks and these will be permanently available upon subsequent logins.
- 2) the administrator of any new Morphonet instance can decide what are the public tracks shown by default to all users.

Sharing the whole user interaction with the dataset, including the view, is however an interesting idea. One might think for instance of the possibility to share user sessions (as the WashU epigenome browser allows), or to record whole sessions of interactions (as Galaxy does). Such sophisticated features will be developed in next releases of Morphonet.

R3.7 "...does not usually include a long-term plan of software maintenance" (line 52). I don't agree with it. I believe the maintenance of all the tool depends on the grant. Many tools keep updating for a higher version until today.

We agree with the referee: the maintenance plan of a scientific research software depends on the grant. What we meant was that it is easier to maintain one specific server instead of a software which needs be installed on different machines, with different operating systems which themselves evolve in time. As a consequence, smaller grants are needed for the maintenance of such tools as MorphoNet.

We have added a sentence in the main text to clarify this point (L53): *"Finally, the development of these tools requires the maintenance of a software which needs be installed on different machines and with different operating systems which themselves evolve in time. This needs a long-term intensive plan of software maintenance, which may not be compatible with the limited resources available in the context of specific research projects, with the risk of rapid obsolescence."*

R3.8 The functions or properties are good, but are relatively simple.

Thank you. We hope to convince you in the following paragraphs that some of the functionalities implemented in Morphonet are actually quite unique and powerful.

Here are just a few examples:

R3.9 It can select the objects one by one, but cannot select the objects in a region. For example, if users would like to select the cells in a certain region which may include 1000 cells, in Meshlab, users could drag the mouse to select the region, then the objects in this region will be selected. But in MorphoNet, users have to select one by one which needs a lot of time and may not be practical. Also, the inside cells are blocked and hard to be selected.

We do not fully agree with the referee here. Specifically, the current version of MorphoNet already has three different ways of selecting collections of objects:

- X+mouse movement selects all objects over which the pointer passes;
- If adjacency information is provided and loaded, the new button "Neighbours" automatically appears and allows the selection of the neighbourhood of selected objects. As such, a few clicks are usually enough to select thousands of cells;
- If any hierarchical group information is provided, collections of objects can be selected from the "Groups" menu (for example, all cells belonging to a given tissue can be selected with one simple click).

We do realize that user needs may vary and that the built-in functions may not satisfy all of them. This is a problem common to all such multidisciplinary and cross-community tools. It is currently difficult for us to foresee what will be missing for each user community. This is why:

1. We set up a feedback system directly from the online interface, to allow users to communicate specific needs
2. The code will be distributed open-source upon acceptance of the article, so that users and developers can complement the tool and share their improvements

R3.10 I think “neighbor” function is a great idea. But based on my test using the data in Fig 2c, “neighbor” function does not work well. The neighbors for a selected object are not the neighbors in the space. There might be some bugs? Or it picks the neighbors with the nearby labels?

We thank the referee for this detailed review. We have checked and identified the problem. As a matter of fact, the function “Neighbours” works well, but for the termite nest dataset (Fig. 2c) the adjacency information provided by the owners of the dataset and implemented in the “Neighbours” function was not correct.

However the referee can test the use of the Neighbours function on the dataset “**Phallusia mammillata embryo (Wild type, live SPIM imaging, stages 8-17)**”, after activating it from the Infos menu.

R3.11 I believe my computer, browser, and internet speeds are all pretty standard. But as testing the tool in about 30 minutes using their prepared data, it got frozen three times with the warning “not responding”. It won’t recover after refreshing. I also tried other browser, Google Chrome, it even does not work. After click the data, it never showed/loaded the data.

We indeed have sometimes observed similar issues during the streaming of datasets in moments of high server load (MorphoNet is currently hosted on a small server at the Research Center for Cell Biology in Montpellier), independently of the MorphoNet user. This issue has now been solved by optimizing the code to improve the platform performances. We have tested the system from different locations, on different OS platforms and it now seems to work seamlessly in most of the cases. We encourage the referee to try again.

In any case, as the code will be released open-source upon acceptance of the paper, users can also install their own private MorphoNet instances on any server.

R3.12 As I mentioned before, there is no “save” property.

If the referee refers here to the possibility to upload selections and annotations made on objects during user interactions with the dataset, this possibility exists: all information can be saved from the “Infos” menu. This is documented in the Help webpage accessible at the url: <http://www.morphonet.crbm.cnrs.fr/HELP/Help.php>. These saved properties will be listed in the Infos menu.

If however the referee refers to the possibility to set a specific set of information and a specific view to be automatically loaded when the dataset is streamed, this is indeed not yet possible via the online interface as explained above.

This is one of the many features we are willing to implement in the next MorphoNet releases.

R3.13 There are some movies showing examples to visualize different data. But I didn’t find the file/tutor to teach how to format and upload the morphological information files.

Explanations and examples on the format for mesh datasets and informations/annotations can be found here: <http://www.morphonet.crbm.cnrs.fr/HELP/HelpFormat.php>

Following the referee’s comments, as explained in Section 1.3 of the **List of main modifications and additions** at the beginning of this letter, we now provide 3 detailed tutorials which illustrate the detailed use of MorphoNet in different contexts.

R3.14 Developing an online tool is hard. We appreciate the work. There are some local tools can processing the data and also have very good package to visualize the data. We expect the online tool could match the properties. I believe this tool will be updated better and better. But I feel the current version is still preliminary.

We thank the referee for his/her thorough analysis of our work. We hope that now, thanks to his/her efforts and to those of the other referees, we produced a far better version of our tool. We wish to point out, however, that MorphoNet does not have the ambition or aim to replace local image analysis and visualisation tools, as its main aim is to share data and properties within and between communities.

To help the creation of user communities and to push the improving process further, we distribute the MorphoNet code open-source to benefit from the contributions of users and developers worldwide for future releases.

Reviewer #4 (Remarks to the Author):

Overall conclusions

R4.1 I'd like to preface my general thoughts by saying that I appreciate the difficulty and complexity of the task undertaken by the authors of this manuscript, namely creating software on the web for scientific use cases that also leverages digital 3D data infrastructure. Succeeding in this task requires not just a deep understanding of the needs for scientific applications but also familiarity and expertise concerning modern standards and practices in software engineering, database design, 3D graphics presentation, and data and metadata modeling. And frankly, typically it is scientists who must stretch their skills from the research domain into other areas to accomplish this goal. As a result, creating tools like this is necessary and valuable work.

We thank the referee for his/her comments. We completely agree with this remark. Due to the requirements of interdisciplinary science and the severe limits imposed by the lack of funding to develop tool-oriented projects, researchers must often cover several competences they may not fully be familiar with.

This is not just our own experience while developing MorphoNet, but above all the reason why we started developing it: to share with interdisciplinary scientists a dedicated and powerful tool to manipulate and navigate within the complex combination of phenotype-genotype data produced by recent advances in life imaging. .

R4.2 The authors of this manuscript describe MorphoNet, a product which seems to be described simultaneously in three ways: 1) as a tool for individual biologists to visualize data from multiple phenomic and genetic lines of evidence to derive new insights; 2) as a service for biologists as a community to archive and share multi-faceted phenomic/genetic 3D data on the authors' instance of MorphoNet; and 3) as a server-level software package which is intended to be distributed to and used by other individuals and institutions as a contribution to the field of digital biological data infrastructure tools. The product as described in this manuscript shows a keen appreciation of a novel and important use case for combining 3D phenomic and genetic data, but at the same time the software as described may not meet modern software development and data modeling standards. It is difficult to fully judge this second point due to a significant amount of scientific jargon and a lack of clear technical language, especially since the manuscript language often confusingly mixes the two fields (i.e., using the opaque term "morpho-browser" to describe an in-browser platform for viewing 3D media and related annotations representing biological data). I also have questions over whether current community work regarding data and metadata modeling was consulted during the creation of this platform. For all of these reasons, I think there is a "risk of rapid obsolescence" for this product, as the authors themselves noted in relation to other tools.

The referee raises here several important points. First, the language employed in the manuscript reflects the fact that this work lies at the border between different disciplines. We have opted for a language oriented towards the designated users of MorphoNet, rather than developers. MorphoNet is indeed an in-browser tool for analyzing 3D media, but in our view it is more than this to the community of potential end-users. It is conceptually a new tool, for exploring a new type of 3D+time morphological data that have never been manipulated massively seamlessly and cooperatively before. We think that such a novelty calls for the use of new terms as "morpho-browser". This term for example purposely parallels forged expressions such as "morpho-space" (referring to the space of all possible forms achievable by a given form model in biology or ecology), "genetic browser" that has been introduced successfully in a recent past to refer to similar type of computational tools to explore genomes or "morpho-dynamics" that has been widely used in recent years by developmental biologists to refer to the study of developing organisms in 3D. As suggested by the reviewer's comment, we thus revisited our text to clarify these new terms when first used.

Second, Morphonet is indeed meant to be used at the three different levels outlined by the reviewer, i.e. as a tool for individual biologists to explore their data, as a service for biological communities to exchange and store their data and as a server tool for institutions to set-up their own data management infrastructure. As for now, these three levels of use have been mostly tested within a limited context of communities and institutions. By opening the software, our aim is ease the possibility to create data-oriented communities that will make use of this tools, exchange their data and good practice. Before and during the software development, we have put a lot of effort in the underlying software engineering approach to optimize the possibility of this software to evolve with time, usage and users communities. We detail this point in the answer to reviewer's comment **R4.4** below.

Coming to the chosen data and metadata structure, since the outset of this project we have discussed and analysed the different possibilities with many researchers working in fields as diverse as computer science, plant and animal biology at different scales, bio- and theoretical physics, image analysis, simulations and archeology. After several rounds of improvement, we have converged on the two formats proposed in the manuscript, one for

morphological data and one for associated metadata (referred to as *information* in the following, in the manuscript and in MorphoNet).

The MorphoNet format for morphological data is built on the standard .obj format, minimally extended to include the temporal links between objects in timestacks and the notion of multi-channel visualisation of data.

The MorphoNet format for *information* (metadata) is inspired by the standard structure of dictionaries in common programming languages. It consists of identifying objects by a vector ID, containing the timepoint of their existence, their numerical ID at the specific timepoint and their associated visualisation channel. This tuple corresponds to the dictionary key. Any property associated with them (within the possibilities implemented as standard information types in MorphoNet) is the associated value. The list of standard information type is:

- time, for temporal links
- space, for spatial links
- group, for grouping objects together.
- float, for any quantitative properties.
- selection, for objects labelling.
- color, for coloring objects.
- string, for any qualitative properties.
- genetic, for boolean or quantitative gene expressions.
- dict, for quantitative information on links between objects.
- sphere, to visualise spheres on the dataset, each associated to a dataset object.
- vector, to visualise line segments on the dataset, each associated to an object.

These formats have been developed, following the many multidisciplinary requirements and needs expressed during the time of the project, to cover the needs for morphodynamic studies.

A more complete description of these formats, together with practical examples, can be found at this url: <http://www.morphonet.crbm.cnrs.fr/HELP/HelpFormat.php>

In this first phase, we had to face with the definition of as simple as possible data and metadata structures, able to cope with a majority of available morpho-dynamical data and with their import and export to other standard data formats. This achieved a first level in the definition of a standardization approach. As suggested by the reviewer, we will need next to move on to a second level in future refinements of these standard structures and meta-data, to converge with web-oriented standards, the definition of standard ontologies and corresponding dedicated mark-up languages (such as the well known SBML for biological processes, cellML for mathematical models in the context of cell systems, and the recent RML for the description of branching system architecture in plants that we contributed to develop [Lobet, G., Pound, M. P., Diener, J., Pradal, C., Draye, X., Godin, C., et al. (2015). Root system markup language: toward a unified root architecture description language. *Plant Physiol*, 167(3), 617–627. <http://doi.org/10.1104/pp.114.253625>] -). In particular, we are part of a world-wide initiative coordinated by the Sainsbury lab in Cambridge, UK, to develop markup languages to exchange multicellular tissue data between research groups, together with models of morphogenesis. We anticipate that the interaction with larger and larger communities of MorphoNet users and developers will allow this format and standards to be constantly optimized. Thanks to the open-source nature of MorphoNet and its foundations rooted in the sharing of data and metadata between users of different disciplines, MorphoNet will largely profit from a large user community that will be able to contribute optimizing MorphoNet standards and guaranteeing its constant development.

This will be eased through the feedback interface we implemented in this first release and that favor an evolution of MorphoNet formats towards increasingly standardized forms.

We edited the main text by adding two sentences in the conclusions on the perspective of MorphoNet development (L201): “*It largely profits from the most recent and well-developed web technologies and introduces a generic strategy for hierarchical representation of biological structure. We expect in the future, thanks to the contribution of large user communities, to develop the MorphoNet data and metadata formats towards standardization.*”

Finally, to even more facilitate the usage of MorphoNet, we have developed two APIs (Python and ImageJ) and several converters, thanks to which other standard formats used for morphological data (multiple formats for 3D image, .vtk and .stl for surface meshes) and for related metadata (notably, csv format) can automatically be converted into the MorphoNet format. These converters can be found at this url: <http://www.morphonet.crbm.cnrs.fr/Converters/Converters.php>

R4.3 To expand on my concerns above, one major omission from this manuscript is whether the MorphoNet software is open source. My suspicion is that it is not, given that the core 3D viewing engine used here, Unity3D, is itself proprietary software. There is certainly a place for proprietary or otherwise closed source software in

certain software domains, such as desktop applications, where individual software applications are relatively independent and often do not interact with each other. But MorphoNet is server software, the authors suggest other users could use their software on other server systems. Server software components rely on being able to interact with other software components, and there is a reason why many server software components, even those produced by for-profit corporations, are open source. If MorphoNet is not open source, this severely hampers any potential for long-term use of this product on other systems.

A relating factor to the open source question is that the authors do not indicate where or if the MorphoNet source code is available. Source code for software of this kind should generally be available unless extenuating circumstances exist, and it is my opinion that scientific web software that is proposed to be run by others on external servers should not be published unless the source code is made freely available and easily discoverable on the web.

Along with this, there is a lack of detail in the manuscript concerning the technical implementation of this software, and what details are present suggest possible concerns.

We thank the referee for this constructive comment. We fully agree with his/her concerns and his/her vision on server-based software requirements. For this reason we have decided to distribute the MorphoNet code open-source with the manuscript.

This code, including both client and server sides and together with all aforementioned tutorials, APIs, converters and with a full documentation and installation procedure, is available on GitLab at <https://gitlab.inria.fr/efaure/MorphoNet>.

R4.4 Software that meets current standards of web development avoids the possibility of obsolescence through being flexible (i.e., able to be displayed through a variety of viewing methods), modular, taking advantage of previously created foundations and frameworks, and using rigorous automated testing. It is difficult to tell whether MorphoNet meets these standards, but if the majority of the application aside from the Unity3D core is custom-written, it is questionable to what degree these standards will be met. These are the benchmarks that determine how difficult it will be for the MorphoNet software to be maintained moving forward or to be integrated into other environments.

The referee raises here another important point. It is true that MorphoNet has not been developed with benchmarks comparable with those of commercial softwares. However, as the referee him/herself states in his/her preface, we have followed user-centered agile software development methods. In this process we have chosen to use specifically high quality, tested and supported softwares (Unity3D, Firefox, Debian), programming languages (C#, PHP, Python), and standard databases such as MySQL.

On top of that, our code is modular and hosted and versioned in the GitLab of the INRIA (French national Institute for research in Informatics and Automation). The INRIA community has a strong experience in scientific software development and maintenance. OpenAlea [Pradal, et al, Functional Plant Biology 2008], a platform developed within the VirtualPlants team at INRIA headed one of the authors of this manuscript, is widely employed by many research groups worldwide and has several millions downloads on gforge. This invaluable expertise strongly supports the benchmarking of MorphoNet.

R4.5 In addition to software development concerns, the limited description given in the manuscript raises the question of whether the authors have consulted and compared with community standards and products when it comes to modeling data organization or creating metadata schema. These are important considerations as they determine whether software systems are capable of “talking” to each other and exchanging data or metadata as required.

This point was addressed in a previous answer (see answer to reviewer’s comment **R4.2**).

In addition, as ImageJ is among the most employed and well-known image visualisation and analysis softwares in the biology community, we developed an ImageJ plugin which permits the direct upload to the MorphoNet server of 3D segmented data and additional related information from ImageJ. Incidentally, On top of that we provide several scripts in Python to automatically convert multiple 3D image formats, .vtk and .stl mesh formats and csv format to the specific data and metadata MorphoNet formats.

R4.6 No mention is made of any other biological digital data web resources (e.g., DataDryad), including those specifically devoted to 3D data (e.g., MorphoSource, MorphoMuseum, Phenome10k).

We have added to our database two new datasets provided by the web resources he/she mentions.

- *Macropus eugenii* (Tamar wallaby) pouch young (Wild type, scan) from Phenome10k;
- *Canariomys bravori* (Fossil, computerized microtomography) from MorphoMuseum;
- There was already a public dataset in the MorphoNet database which comes from DataDryad: *Cascolus ravitis* (Fossil, digitally captured images).

Finally, we contacted the curators of MorphoSource, but we have not been given access to their data.

We now also refer to these resources in the main text (L58): *“Web-based morphological databases exist (MorphoSource, Phenome10k, MorphoMuseum, DataDryad), some of which come together with an online data visualisation interface. However, the set of user interactions with the dataset is much more limited, and no additional information can be uploaded to and/or projected onto the dataset.”*

We do believe that one of the next important points to address in the next release is the direct connection of MorphoNet tools with these web databases through dedicated APIs.

R4.7 More concerning is that the authors propose a “universal strategy to hierarchically group objects” but do not discuss or seem to account for the significant amount of effort that has already been committed by working groups creating standardized schemas for data and metadata modeling (e.g., PREMIS, PROV, Dublin Core, Darwin Core, this could be a very long list).

Is the “universal strategy” suggested here compatible with linked data platforms or the Resource Description Framework?

The reviewer is perfectly right in pointing out this issue. Our usage of the term “universal” was a bit misleading here. What we had in mind was rather a problem related to the modeling of biological structures as multi-scale systems in a universal manner and the possibility to express this using hierarchies of topological and geometric objects, such as cellular complexes. This in itself is a challenging issue for which we provide a first tentative approach to define generic data-structures to represent forms in development.

In the future, as mentioned by the reviewer, this approach must be refined and complemented by a more generic data oriented modeling standardization that encompasses efforts made in the languages defined for expressing universal semantics such as RDF or its variant N3. This will require the extra effort to define and agree on standard ontologies for our domain, that is for the moment completely lacking. The availability of a tool such as Morphonet will ease the federation of such efforts.

We thus modified our sentence, which now reads (L111): *“We likewise developed a strategy to describe biological structures as multi-scale systems and to express this using hierarchies of topological and geometric objects, such as cellular complexes.”*

R4.8 Creating successful software projects in the current rapidly evolving time period requires integration with previously existing work, as well as building tools in such a manner as to make them more easily integrable with tools to come. A lack of attention to these factors is the underlying reason why much scientific software experiences obsolescence at such a fast rate. As an example of scientific software done correctly, strong consideration for these concepts is why the use of programming languages like R for statistical analyses has been so revolutionary for the practice of science in the last decade. For a concept like MorphoNet to be successful as a digital infrastructure tool, its creators should similarly heed these principles to avoid the same end they themselves observed that many scientific software products meet.

We share the same vision of the referee’s on important characteristics required for scientific softwares to be successful, as exemplified by our previous experiences on large scale scientific software development AMAPmod [Godin and Guédon 2002], (3D Virtual Embryo [Tassy, et al. Curr Biology 2006], OpenAlea [Pradal, et al, Functional Plant Biology 2008] , MARS [Fernandez, et al Nature Methods 2010], LPy [Boudon et al., 2012], MovIT [Faure, et al, Nature Communications 2016], Aniseed [Brozovic, et al. Nucleic Acid Research 2017]], ASTEC [Guignard, et al. bioRxiv 2017])

We believe MorphoNet meets both requirements highlighted by the referee.

There are four main points we want to stress that play in favour of a high MorphoNet potential for dissemination and to fight efficiently against obsolescence drift:

1. MorphoNet’s code is open source. This will allow users and developers to constantly improve the tool.

2. MorphoNet's dedicated formats which facilitate data and metadata sharing. In addition, we exemplify the integrability of MorphoNet with other tools and formats by providing scripts to automatically convert multiple file formats to the native formats of MorphoNet. This strategy, which can be developed in time, easily allows the interaction of MorphoNet with many other standardly employed tools.
3. The use of API to interact with external tools or databases. The Python API allows users to employ the computational power of Python for producing morphological information directly uploaded to MorphoNet. The ImageJ API which creates a plug-in in the ImageJ application, easy to use even for users without any computer-science experience. The genetic API, which permits to parse genetic databases and to integrate the extracted information as genetic information on MorphoNet. The use of the latter is exemplified by the integration of the ascidian genetic database ANISEED on the MorphoNet server.
4. The development of the software based on common and used technologies with a large community such as Unity3D, Firefox and WebGL.
5. We already identified a relatively large community of interested users in both plant and animal research areas, which will use MorphoNet and contribute to its dissemination right from the beginning. We also have joint projects with a few other labs who will contribute directly to MorphoNet development.

Other General Comments

R4.9 Trying out the platform, I seem to be unable to get surface models to rotate in the viewer? Additionally, the layout of the website seems to be not working correctly in Chrome V68.0.3440.106 on OSX Sierra, with the viewer overlapping the header bar with the site logo. Also, on OSX Retina displays, the UI seems very blurry.

We optimized MorphoNet performances on Firefox since it is one of the most used free web browser. We do not yet assure, for now, a high-quality experience when using other browsers.

We have tested the tool on several operating systems: Windows (tested on Windows 10), Linux (tested on Ubuntu 14.04, 16.04, 18.04), OSX (tested on El Capitan 10.11, Sierra 10.12, High Sierra 10.13 and Mojave 10.14) and on Firefox (versions 60 to 63), with at least 8 GB of RAM.

About the image quality issue the referee reports on OSX Retina, we apologize but we have never experience any such problems, although we use the tool everyday on OSX Retina displays. This could be related to the graphic card used by the referee. The minimal requirements on the graphic card for a smooth MorphoNet experience will be addressed as soon as possible.

We hope that the open source distribution of the code will help optimizing MorphoNet performances on other web browsers. In order to receive and address users comments we have put in place a feedback system directly from the online interface.

R4.10 The name MorphoNet seems to be already used: https://link.springer.com/chapter/10.1007/978-3-642-15754-7_72

We thank the referee for pointing out this. We are aware of this paper, but it does not refer to any softwares or platforms and, above all, the name MorphoNet is not protected by copyright.

R4.11 Where is the source code? This is using Unity3D engine, so is this not then open source? Open source software is crucial for collaborative development of scientific software, so a visualization engine wrapped around Unity3D seems like will be very difficult to keep maintained. At the very least, it removes the potential of this software to be incorporated into other projects, which means it cannot be considered a contribution to the evolving field of digital data infrastructure resources.

The source code is provided in the INRIA GitLab (<https://gitlab.inria.fr/efaure/MorphoNet>) as mentioned in the new version of the supplementary information. The Unity3D license for public research allows the free distribution and integration of Unity3D-wrapped engines for public, non-profit uses and in particular for scientific research. Before turning to Unity3D we have explored open source alternatives exploiting the WebGL library. None of these solutions is comparable to the features of Unity3D needed for our project. Especially in user-centered development process, Unity3D offers a very intuitive interface for interaction with complex 3D environments.

That being said, we completely agree that Unity3D is a private company software and we do believe that, once open-source libraries will match the features needed for MorphoNet, it will be a wise choice to turn to them. The modular architecture of MorphoNet, keeping separated Unity features from independent MorphoNet ones, should make it a relatively easy task if necessary.

R4.12 There is a fair amount of novel scientific jargon in the manuscript, as well as a general unclear mixing of scientific and technical terms and vocabularies. I am a trained morphologist and a software developer, and so I believe I should be in the audience to understand a manuscript that involves both of these fields. However, I find the language simultaneously too vague and too jargon-heavy (i.e., terms like “morpho-browser” or “web-based morphodynamic browser”). In terms of what I mean by confusion, this manuscript discusses several things: the basic capabilities of the software components the authors use (WebGL being able to load 3D graphics in a browser), what the authors have used these components to achieve (to create an in-browser 3D viewer capable of showing multiple mesh objects and coloring them), and how these tools are meant to be applied (importing data relating to different lines of evidence and visualizing them together for better understanding). But these different levels are not clearly discussed, and the explanations provided often span all of these levels in an unclear fashion.

There are several levels of complexity depth in the work presented in the manuscript, mixing computer science, the needs of scientific research and the will for our tool to be accessible by many non-experienced users. As such, technical details in the presentation of the main ideas are avoided to try to achieve a more global vision on our result. We have given in the main text many basic ideas pertaining each of these levels of complexity, in such a way that experts in each related field can grasp the motivations behind our work. The audience of our manuscript is however the average MorphoNet user, not developers. The computer science expert will find all technical details in the documentation of the MorphoNet GitLab (<https://gitlab.inria.fr/efaure/MorphoNet>). The interested user will benefit from the detailed tutorials now available online (http://www.morphonet.crbm.cnrs.fr/Tutorials/MorphoNet_Tutorials.php). The researcher interested in quantitative calculations and data-inspired morphodynamic models will profit of the use of the Python API (<http://www.morphonet.crbm.cnrs.fr/HELP/HelpAPI.php>). The main text gives a general overview of the spirit used in developing MorphoNet, which mixes all these layers. Such a homogenization effort has been our choice to better communicate the multidisciplinary relevance of our work. We note that the 4 other reviewers appreciated the clarity and the presentation style of the manuscript.

We agree with the referee on the fact that some terms in the main manuscripts need clarification. Notably, the term morphodynamic browser is not well characterized in the manuscript. To clarify what we mean by it, we have modified the last paragraph (L63), which now reads: *“The open-source web-based tool presented here (<http://www.morphonet.crbm.cnrs.fr>) shows that the exploration and analysis of diverse large-scale imaging datasets can be used for a conceptually analogous philosophy to that which presided over the development of generic web-based genome browsers. MorphoNet allows the interactive visualization and sharing of complex morphodynamic datasets, onto which quantitative and qualitative information can be projected. Central to the concept is the definition of a unified, human-readable data format. In this sense, one could refer to MorphoNet as a morphodynamic browser.”*

R4.13 Who is the audience of this paper? Is it to advertise the authors’ instance of this platform to scientific users who might use their instance? Or is it to suggest that other bioinformaticians could either set up their own instance or use the components of this software to create new tools? There is not enough technical depth here to address either of the last two audiences, but the authors do seem to want to be writing for them.

MorphoNet is targeted to various different audience communities by its very own interdisciplinary nature. However, as already commented, the main audience of the paper is the MorphoNet scientific end-user. On the other hand, the complete technical information available on the MorphoNet GitLab (<https://gitlab.inria.fr/efaure/MorphoNet>) is for those who would like to set up their own private MorphoNet instance.

To better guide the user of MorphoNet we further provide three detailed Tutorials (http://www.morphonet.crbm.cnrs.fr/Tutorials/MorphoNet_Tutorials.php) showing the use of our platform in investigating specific scientific questions.

Specific Line Comments

R4.14 43: There is no discussion here of the difference between “visualization tools” and biological databases and platforms to share data, and the only previous works discussed seems to be to local software? No mention of online databases that provide similar functionality (MorphoSource, Phenome10k, MorphoMuseum) or general online databases to share scientific data such as DataDryad, etc. Not even Sketchfab and its annotation capabilities? This does not seem like an adequate characterization of the field as it currently stands.

We thank the referee for pointing out these sources. Due to the strong limitations in the number of references imposed by the article type and the fact that these are simply visualisation tools with limited interaction possibility, we needed to select only a few of the many references worth mentioning. We have added references to these tools and a sentence in the main text where we compare our platform to them in the new paragraph (L58): *“Web-based image databases exist (MorphoSource, Phenome10k, MorphoMuseum, DataDryad), some of which come together with an online data visualisation interface. However, the set of user interactions with the dataset is much more limited than what can be found in installation-based platforms, and no additional information can be uploaded to and/or projected onto the dataset.”*

Moreover we now also provide some datasets in MorphoNet downloaded from the sources mentioned by the referee.

R4.15 69, 71: This is a very high level description of the implementation of this platform, I would have liked to see much more technical detail. Also again, open source?

All details and the open source code are now given in the GitLab at <https://gitlab.inria.fr/efaure/MorphoNet>. We have added this information in the main text at the end of the paragraph the referee refers to with the sentence (L84): *“The whole open source code of MorphoNet and related documentation are available on a GitLab at <https://gitlab.inria.fr/efaure/MorphoNet>”*

R4.16 79: I appreciate and respect the broad scope of the application this software is meant for. That being said, this statement oversells the fact that the authors have implemented a 3D surface mesh in-browser web viewer (of which there are other open-source examples, such as three.js), and can therefore display 3D surface meshes representing many types of biological surfaces. This tool could theoretically also display 3D surface meshes representing buildings or hammers or space shuttles, for instance. More language should be provided to describe exactly how this tool supports a broad range of morphological domains that is not simply encapsulated by the idea that a mesh viewer can display mesh data.

We agree with the referee on the fact that many visualisation tools for meshes are available on the web, as he/she points out several times. As said already, these tools are only simple mesh visualisation platforms without the possibility to create, save, upload and share associated information and quantitative properties. Also, most of these tools are specific for one or few research communities.

The specificity of MorphoNet is not its ability to visualise surface meshes, but rather its flexible data-metadata connection, which provides invaluable help to quantitative multidisciplinary studies of morphogenesis and developmental biology. On top of that, MorphoNet offers a sharing and linking environment to different research communities, which can compare results and data through a common format-driven language.

We have shown that our platform can efficiently be used to visualise and interact with datasets coming from widely different scientific communities.

R4.17 100: Did the authors reference any of the many best-practice metadata or data organization standards created by standards working groups? PREMIS, PROV, Linked data models, Resource Description Framework, etc.?

Please refer to our answer to reviewer's comment **R4.7**

R4.18 101: Does the sentences on this line and following it for the rest of the paragraph essentially boil down the fact that the tool allows display of multiple meshes, with turning visibility on and off per mesh?

Indeed, this was not our message. What we mean is that there is more than just visualisation of multiple meshes involved. Objects in different meshes can be linked, meaning that users interactions on one object in one mesh channel is automatically reported to the linked object in the other mesh channel. For instance, if one mesh channel represents cells membranes and the second is for cells nuclei, coloring a cell in one channel will automatically color the same cell in the other. To clarify this point we have added the following sentence in the paragraph mentioned by the referee (L115): *“Importantly, objects in different visualisation channels can be synchronized and actions performed on one channels can be automatically reported on linked objects on other channels.”*

R4.19 Paragraph 116: In terms often used by the image viewer community, these are annotations, though this word is not used by the authors and no comparison is made to other implementations of annotations. How does this work compare with other current effort to create standardized annotations? Are the annotations created here comparable with other tools?

We agree that “annotations” is a standard term used by the image viewer community, and often refers to manually-curated metadata. However, what we deal with in MorphoNet is a more general concept of data-centered information, projected onto elementary objects. As such we decided not to use the term “annotations” in order to provide more flexibility and generality in the choice of the property to be represented, such as time links, genetic expressions, group information.

As commented before, this format is compatible with other standard properties format (such as csv) thanks to the automatic converters (<http://www.morphonet.crbm.cnrs.fr/Converters/Converters.php>) and the APIs (<http://www.morphonet.crbm.cnrs.fr/HELP/HelpAPI.php>) we developed.

R4.20 124: Does this mean this tool is capable of reading a variety of data formats or data structures that relate to genomic expression data from RNAseq, etc.? If so, what are these formats or standards? Are these all data standards particular to this project?

MorphoNet is able to read its own specific format for genetic information. Such format is flexible enough to include several types of data, coming from different experimental techniques. However one must always convert her acquired data into the MorphoNet format before uploading them, which can be done via python API (<http://www.morphonet.crbm.cnrs.fr/HELP/HelpAPI.php>).

We did not have yet the opportunity to work with RNAseq data associated with our morphological datasets. If the case arises, however, we will produce and provide an automatic converter also for this kind of data.

R4.21 128: What databases are connected via APIs? What are the “universally formatted files”?

Until now we have only connected the genetic database ANISEED, through the Aniseed API and a Python converter. In general, any genetic expression data, be it taken from an external database or a manually-entered one, must be formatted into the standard MorphoNet genetic format described in the Methods.

We now explicitly mention ANISEED in the sentence.

We agree that “universally formatted files” is a misleading expression. We have replaced this expression to clarify the meaning of the sentence. It now goes as (L146: “[...] in the dedicated genetic MorphoNet format, detailed in the section Morphological Information in the Methods.”

R4.22 283: This sentence describes a fundamental model for all web resources, I think the explanation should be more elaborate to connect this statement with the specific tools discussed.

The referee is right, the sentence was just a generic introduction to the specifics of the web architecture. We have added more details that makes a better connection with the specific features of MorphoNet. The sentence now reads (L337): *“MorphoNet is a web-based application composed of a front-end application on the client (internet browser) and a back-end support on the server which communicate through classical http requests.*

The structure of MorphoNet integrates a database for uploading and sharing, which is hosted on the back-end, and a 3D interactive interface for visualisation and other advanced dataset interaction, which is hosted on the front-end.”

R4.23 286: Asking users to specifically use one browser does not reflect modern web development standards

We agree with the referee on this point. However we wish to point out that for the first delivery of MorphoNet we have chosen one of the most used free internet browsers. We wish however to point out that, to our non-extensive tests, the MorphoNet application responds well on other browsers such as Chrome and Safari. One known problem with Chrome is for instance the security level imposed by Google, which interferes with the upload/download of information via the online interface. We hope, thanks to the contributions of users and developers and to the open-source distribution of the code, to soon be able to optimise MorphoNet for other browsers in the next releases.

R4.24 287: This is an unusual request for users (and is also not explained how this is done), given that other websites displaying 500MB meshes such as Sketchfab and MorphoSource have no similar requirement

The increase of browser cache mentioned here is not to be read as due to a technical limitation, but rather a suggestion for people who need to stream several times many datasets. Having a larger cache size would largely reduce the time required for repeated streamings. Fortunately, with the last Firefox update, the cache limitation does not exist anymore, hence we remove this sentence from the text.

In the second part of this paragraph, we mentioned the limitation memory of 2GB due to the WebGL structure: for visualization only, Sketchfab and MorphoSource have actually the same WebGL memory limitation as MorphoNet. However, MorphoNet offers more interactivity with the dataset such as superimposed quantitative and qualitative information, which requires additional memory allocation. The sentence was not clearly written, and we have now clarified it by adding some new information and modifying a sentence in the "Implementation" section of the Methods (L346): *"The main technical limitation of MorphoNet, imposed by the very structure of WebGL, is the 2GB limit of memory use, as is the case for many online mesh visualisation platforms. However, for a better interactivity with the dataset during more advanced user interactions (information upload, computations, lineage tree exploration), we suggest to limit the total size of a dataset mesh to 500MB. This can be achieved by decimating the dataset mesh, i.e. decreasing the number of polygons composing the mesh, for instance through MeshLab (<http://www.meshlab.net>). In the same spirit, in order to optimize the interaction with the dataset, we recommend to limit the total number of elementary objects in each dataset to half a million."*

R4.25 291: Is this section a description of the implementation of the website or a guide for how to import data into the tool? The narrative of these paragraphs is confusing

The referee is right, this paragraph was ambiguous, and we have modified it as explained in the answer to his/her previous remark. As a matter of fact, we were only raising a word of caution for MorphoNet users when preparing their datasets for upload. The specific upload process is now explained in details in the three Tutorials we provide online (http://www.morphonet.crbm.cnrs.fr/Tutorials/MorphoNet_Tutorials.php).

R4.26 300: And yet the authors recommend only using Firefox?

Yes we do and, as the referee evokes in his/her preface, to cover every task in the development of such multidisciplinary tool is cumbersome for just scientists. So as the first release of MorphoNet we recommend, but not constrain, to use a free web browser accessible on any computer.

R4.27 310 (whole paragraph): A general issue here is that this website (without logging in) does not seem to provide any evidence for the technical claims regarding any of its functionality, including the API. There is also no publicly accessible documentation.

The referee is right, and this is only a temporary solution: a free, public access to open datasets, API and documentation will be granted as soon as the work is published.

R4.28 Also, can the software have a GNU license when using Unity3D as a core component?

Yes, it is possible to have a GNU license even when using Unity3D. We have in particular used a GNU GPL license, named CeCILL, provided by our government employer (<http://www.morphonet.crbm.cnrs.fr/Dev/license.php>). What we are licensing is only the C# code that exploits the Unity3D core.

R4.29 391: Most common mesh format is .obj? It is certainly a common file format, but I imagine something like .stl is more common.

We have changed the sentence to a less strong meaning: obj is indeed maybe not the most common, but surely one of the most used formats for meshes. We have changed the sentence (L448) to *"we employ one of the most used format"*

In addition, we now provide some scripts (<http://www.morphonet.crbm.cnrs.fr/Converters/Converters.php>) to automatic convert .vtk and .stl mesh formats to the MorphoNet .obj format.

Reviewer #5 (Remarks to the Author):

I am very excited to start using MorphoNet! I have helped to create some browser-based biomedical image viewing interfaces, and am used to using the ascii VTK format and visualizing with ParaView, Meshlab, and other interfaces that do not permit easy online viewing, combining, and sharing. The authors deserve particular credit for making MorphoNet open source, online, and free, with open, widely used standards for its format and API. I was excited to watch all of the videos to get a practical sense not just of MorphoNet's features, but of the practical manner in which a user interacts with the front-end.

R5.1 I look forward to converting my own (.nii and .vtk) brain imaging data and brain image analysis software (mindboggle.info) output to .obj to enable people to explore with MorphoNet. I would also be interested to have MorphoNet host these public data for use by the scientific community, and hope many others with large datasets will do the same.

We thank the referee for his enthusiastic comments.

Three suggestions:

R5.2 (1) Please provide guidance for a reader/user to reformat (e.g., VTK) surface mesh data (you mention Meshlab) or to create .obj meshes from (e.g., dicom, nifti, stl) volumetric data for use in MorphoNet.

We followed the referee's suggestion and now provide here detailed explanation on how to use our dedicated converter to transform other image or mesh formats into the MorphoNet data format. In addition, following reviewer's suggestion, we now provide automatic converters from standard .vtk and .nii to .obj format. Converters can be found here: <http://www.morphonet.crbm.cnrs.fr/Dev/Converters/Converters.php>. They will produce obj files, already in the MorphoNet format, ready to be uploaded to the MorphoNet server.

As mentioned in Section 1.3 of the **List of main modifications and additions** at the beginning of this letter, we also provide now three detailed tutorials on the use of MorphoNet. Tutorial 1 also provides a practical example of the use of these converters.

R5.3 (2) lines 217-218: "The dataset position and orientation in space can be reset to default values, and the default orientation can be set (only by dataset owner)."

It would be very nice to capture a view of a dataset (orientation, crop, zoom, and any modifications to colors, transparency, etc.) and share that view as a starting point for another user (more than just as a screenshot).

This is a very interesting suggestion, which we will implement as a priority in producing the next MorphoNet release.

R5.4 (3) I appreciate the dynamic changes in color-coding and transparency of objects. Would it be possible to view the meshes or surface normals, or to cull back surfaces of objects?

We tried to implement this suggestion during the revision time. For this, we have tested several mesh-specific shaders and, although they all work fine locally, they all conflict with the WebGL rendering. We are working to solve this issue, and we hope this option will be implemented in the next release of MorphoNet.

R5.5 It would also be nice to have a static view with 3-D quiver plot to contextualize movement/change between frames.□

Among the different standard information types available to the MorphoNet user there is the possibility to upload line segments (type:vector). Users need to specify the initial and final point of the segment and the size of the line at each of these two points. In this way one can easily represent quiver lines contextualizing movement, normal directions or anything else one may think of that can be represented as a line or a vector. The calculation of these properties to be represented as lines will not be done on the MorphoNet server in order not to alter the

fluidity of the MorphoNet experience. However, thanks to the Python API, one can easily calculate and upload such information.

As an example, the referee can now find in the dataset "**Phallusia mammillata embryo (Wild type, live SPIM imaging, stages 8-17)**" the information called "Quiver Lines", which represents the normals to the external surface of all embryonic cells at several timepoints.

R5.6 Edits:

- * 17 biology, at scales ranging from the molecule to the functional organ. To support this big-data
- * 18 revolution, we have developed a concept of generic web-based morphodynamic browser to
- * 26 to the development -> with the development
- * 27 specimen -> specimens
- * 34 need of -> need for
- * 71 clouded -> cloud-[based, stored, accessed,...]
- * 90 chambers volume -> chamber's volume
- * 91 Human -> human
- * 99 universally-formatted files, that -> universally formatted files that
- * 171, 173, 176 featuring -> features
- * 178 discussions on -> discussions of
- * 213 numerical id -> numerical ID
- * 224, 226, 231, 234, 363, 405, 444, 451, 463, 470, 479 associated to -> associated with
- * 224 Information are -> Information is
- * 225 are color labelling applied -> are color [labels applied, label applications]
- * 240 downloaded in as -> downloaded as
- * 243 Specific objects interactions -> Specific object interactions
- * 245 cluster group -> cluster a group
- * 253 expression on data is -> expression data are
- * 269 as .mp4 -> as an .mp4
- * 283 composed by -> composed of
- * 286 on any Internet browsers -> on any Internet browser
- * 293 dataset mesh -> a dataset mesh
- * 300 codes -> code
- * 301 browsers -> browser
- * 303 on application/request -> by [request, application/request]
- * 306 etc..). -> etc.).
- * 314 users -> user
- * 317 data is uploaded on -> data are uploaded to
- * 318 each new meshes in -> each new mesh to
- * Table 1. MorphoNet Users Rights. -> MorphoNet User Rights.
- * 345 to MorphoNet -> to the MorphoNet
- * 357 nor -> or
- * 372. a public -> public
- * 379 advantage from -> advantage of
- * 381 his -> its
- * 385 the id -> the ID
- * 390, 463 under -> in
- * 465 overlapped to -> superimposed on

Supplementary:

- * color coded -> color-coded
- * names, corresponding to each organ, is -> names, corresponding to each organ, are
- * along to different -> along different
- * three dimensional -> three-dimensional
- * VTK library -> the VTK library

We thank the referee for his thorough analysis of the manuscript and for his syntactic corrections. We have modified the manuscript throughout accordingly.

Cheers,

Arno Klein

Reviewers' comments:

Reviewer #2 (Remarks to the Author):

In this revised version, the authors have effectively addressed the reviewers' comments. Specifically, with the addition of tutorials, specific information in the text, such as the integration with Fiji and Python, which is quite appealing, and new public datasets illustrating the features of MorphoNet.

The authors should once more be commended for this important and unique work.

Reviewer #3 (Remarks to the Author):

This is the 2nd round review from reviewer #3.

In this updated version, the tool does get improved and address or answer most of my concerns. There are still some minor comments are not fully addressed. But I think, overall, this version should be interested and benefit for the community.

For some of my questions, the authors decided to develop in the next release. For example, "the referee is right that Morphonet users cannot currently save a specific view on the dataset and return to this view upon login. This feature will be implemented in the next platform release" and "Sharing the whole user interaction with the dataset... such sophisticated features will be developed in next release of Morphonet."

The current version of MorphoNet has three different ways of selecting collections of objects. However, X+mouse movement can only select regular region perpendicular with x, y, or z axis. The other two selections can be achieved only in the case that adjacency information or hierarchical group information has been provided. What I suggested is that user can simply use the mouse to draw a region and all the objects located in the region can be selected. I feel this is a convenient feature. Unfortunately, the authors did not take it for current version. Hope they can consider to develop it in the next release.

For the problem that my PC got frozen. I don't have this problem now. IE browser can load the data, but Google Chrome still cannot load the data.

Reviewer #4 (Remarks to the Author):

The authors of this manuscript have very clearly laid out what they have done to address reviewer comments and suggestions. Their decision to release the MorphoNet source code as open source is excellent to hear, and it will greatly increase the potential for success of this software. Similarly, the addition of APIs, external tools, and tutorial videos are also appreciated and will increase the usability of this product. I believe that MorphoNet as explained in this revision makes for an excellent tool on the level of use by individual researchers for integrative multi-disciplinary research. That being said, I continue to have serious concerns about MorphoNet as a service through which researchers will share data with each other and/or for public educational outreach. I will go into further detail below.

As a tool for end-user researchers to visualize morphological data with other lines of evidence, I think that MorphoNet has the potential to be very useful for doing integrative multi-disciplinary research. Making the source code open source and adding additional tools have largely addressed my concerns here, since the digital infrastructure behind MorphoNet can now be transparently examined by the community.

The additional public datasets that have been added to MorphoNet are an interesting contribution. They certainly are enjoyable to peruse, and they expand the diversity of data available to users. They do inspire some questions that I think are relevant to consider. For one, I'm not sure why datasets were taken from MorphoMuseum and Phenome10k but not MorphoSource, since MorphoSource has thousands of datasets that are public access just like those from MorphoMuseum and Phenome10k. The authors state that "we contacted the curators of

MorphoSource, but we have not been given access to their data." But MorphoSource has no centralized curators that determine access or lack thereof to data on that repository platform, as it is a platform designed to provide individual data uploaders the tools to control access to their own data. Since digital data is relatively novel, special care must be taken to allow for highly variable preferences by data contributors regarding the restricting of access and discovery. I'm not sure whether and how MorphoNet addresses this issue, but in any case there are many, many open access datasets on MorphoSource they could potentially use.

Additionally, the practice of "reaccessioning" datasets from one digital 3D repository into another gives me pause. The ideal is of course for digital data to be easily available across distributed formats and sources. But if (for example) one repository encourages tracking of data usage and datasets from that repository are mirrored in a second repository that does not do similar tracking, does this dilute the quantitative evidence for data use and make users less likely to share digital data? There are likely ways to address this issue, but it has not been strongly addressed here. And I do recognize that this is not the "point" of MorphoNet per se, but I feel the reaccessioning datasets from other repositories raises some of these questions.

Services for data sharing should ideally be built in such a way as to increase the traceability of data and to link it with both the physical specimens from which the digital data derives and any original or secondary sources from which digital data is gathered. MorphoNet does provide links per dataset to two sources: the source of digital data and a cited publication. This does provide an example of a way to trace data, and the sourcing is very clear. But MorphoNet does not seem to include in its data model any facility for explicitly categorizing datasets according to the physical objects from which digital data derives. Rather, MorphoNet seems to organize its media into datasets, where each dataset can have some information regarding the biological specimen it represents. How does this model interact with the case where for a single specimen, multiple datasets have been produced? It seems these would be listed as two entirely separate datasets.

Additionally, the higher level organization of datasets seems to be more or less arbitrary. One dataset is sorted into strictly taxonomic categories such as "Ascidian -> Phallusia mammillata", while another is sorted using taxonomic and topical categories such as "Human -> Anatomy", and yet another is sorted using taxonomic and paleontological categories such as "Fossil -> Crustacean". Certainly all of this metadata (and I do think that metadata is a more explicit and specifically recognized term than "information" and would strongly suggest the authors consider adopting it) should be tracked, but placing it all together in a single system of hierarchical categories divorced from context is only likely to confuse rather than to assist in data discovery and access. This is the basic gist I'm getting at: that the basic structure of MorphoNet makes it an excellent tool for a small number of datasets when used by a few individuals, but significant problems are likely to be encountered with any significant scaling.

In conclusion, I feel that MorphoNet has significant potential to be extremely useful for a subset of the use cases that the authors have proposed for this tool, that being use by an individual for their own scientific research. Regarding other proposed use cases, such as a service for data sharing, data archiving, or as server-side software to be deployed by technically inclined end users, I feel that MorphoNet is of limited utility and requires significant further development and rethinking before it can serve those aims.

List of main modifications and additions

1. Asked by reviewer 3

a. Drag-and-select.

We have implemented the drag-and-select feature mentioned by the reviewer. Users can now draw a region with their mouse and select all objects falling within it

b. Google Chrome issue.

We tried to reproduce the issue encountered by the reviewer, by testing MorphoNet on two of the last versions of Chrome (72.0.3626.121 and 73.0.3683.75) on Windows 7, 8, 10, on MacOS Sierra 10.13, Mojave 10.14 and on Ubuntu 14, 16, 18. All our tests could not reproduce the problem the referee experienced. To the best of our tests, MorphoNet is thus fully compatible with the latest Chrome browser versions. This is now mentioned in the Implementation section of the Methods (lines 356), by the following sentence:

"We also extensively tested MorphoNet on two versions of Chrome (72.0.3626.121 and 73.0.3683.75) on Windows 7, 8, 10, on MacOS Sierra 10.13, Mojave 10.14 and on Ubuntu 14, 16, 18. MorphoNet is fully compatible with the tested Chrome browser versions."

2. Asked by reviewer 4

a. Integration of a dataset from MorphoSource.

In response to the reviewer's suggestion we now show, as a further proof of principle, that a computer tomography dataset from MorphoSource (a *Human* bone dagger) can be integrated into MorphoNet.

b. Reaccessioning issue.

To facilitate the traceability of the original data source, we now provide, when applicable, specimen ID and original database information in the metadata of each dataset imported in MorphoNet from public databases or online repositories.

While the solution of the reaccessioning issue is way out of the scope of our work, we added two sentences in the main text to draw attention to the problem and to invite users to cite both MorphoNet and the original source. These sentences are in lines 207 and read:

"Due to the lack of unified formats for morphological data and metadata, uploading a dataset from an external database in Morphonet leads to the duplication of this dataset in a slightly different format. To reduce the impact of this reaccessioning issue, we invite Morphonet users to always provide the original dataset ID in the metadata of the uploaded copy, and to cite both MorphoNet and the original database and ID in scientific publications".

c. Dataset organisation.

We now organise datasets in MorphoNet based on their taxonomy, as defined by the NCBI taxonomy database (<https://www.ncbi.nlm.nih.gov/Taxonomy/taxonomyhome.html>). It is also possible to label each dataset, according to its origin, as "Observed" (i.e., created from experimental observation), "Simulated" (i.e., created *in silico* according to an underlying mathematical/computational model) or "Drawing" (i.e., created *in silico* by human-supervised drawing).

Detailed answer to reviewers' remarks

Reviewer #2 (Remarks to the Author):

In this revised version, the authors have effectively addressed the reviewers' comments. Specifically, with the addition of tutorials, specific information in the text, such as the integration with Fiji and Python, which is quite appealing, and new public datasets illustrating the features of MorphoNet. The authors should once more be commended for this important and unique work.

We thank the reviewer for his/her support and positive opinion on our work.

Reviewer #3 (Remarks to the Author):

In this updated version, the tool does get improved and address or answer most of my concerns. There are still some minor comments are not fully addressed. But I think, overall, this version should be interested and benefit for the community.

For some of my questions, the authors decided to develop in the next release. For example, “the referee is right that MorphoNet users cannot currently save a specific view on the dataset and return to this view upon login. This feature will be implemented in the next platform release” and “Sharing the whole user interaction with the dataset... such sophisticated features will be developed in next release of MorphoNet.”

The current version of MorphoNet has three different ways of selecting collections of objects. However, X+mouse movement can only select regular region perpendicular with x, y, or z axis. The other two selections can be achieved only in the case that adjacency information or hierarchical group information has been provided. What I suggested is that user can simply use the mouse to draw a region and all the objects located in the region can be selected. I feel this is a convenient feature. Unfortunately, the authors did not take it for current version. Hope they can consider to develop it in the next release.

For the problem that my PC got frozen. I don't have this problem now. IE browser can load the data, but Google Chrome still cannot load the data.

We thank the reviewer for his/her comments and his/her overall positive opinion on MorphoNet. As a matter of fact, as the referee points out, we needed to prioritize the development of some features and leave some others out of the present version, postponing their development to the next MorphoNet release. However, in light of the point raised here by the referee, we reconsidered the priority list.

In particular, we have now implemented the drag-and-select option he/she suggested. Users can now use this functionality for multiple object selections, in addition of the ones already available before. This feature, together with all other interaction shortcuts, is shown and explained in the shortcut menu at the top-left corner of the visual interface.

We have also looked into the Google Chrome problem mentioned by the referee. We tested MorphoNet on two of the last versions of Chrome (72.0.3626.121 and 73.0.3683.75) on Windows 7, 8, 10, on MacOS Sierra 10.13, Mojave 10.14 and on Ubuntu 14, 16, 18. All our tests could not reproduce the issue the referee experienced. To the best of our tests, MorphoNet is thus fully compatible with the latest Chrome browser versions. This is now mentioned in the Implementation section of the Methods (lines 356), by the following sentence:

“We also extensively tested MorphoNet on two versions of Chrome (72.0.3626.121 and 73.0.3683.75) on Windows 7, 8, 10, on MacOS Sierra 10.13, Mojave 10.14 and on Ubuntu 14, 16, 18. MorphoNet is fully compatible with the tested Chrome browser versions.”

As we experienced a few problems with older versions of Chrome, the referee may have used one of these versions. If, the referee keeps experiencing this issue, he/she should send us a user feedback with the specifics of his/her OS, machine and Chrome version, to provide enough information for us to work out the problem.

Reviewer #4 (Remarks to the Author):

The authors of this manuscript have very clearly laid out what they have done to address reviewer comments and suggestions. Their decision to release the MorphoNet source code as open source is excellent to hear, and it will greatly increase the potential for success of this software. Similarly, the addition of APIs, external tools, and tutorial videos are also appreciated and will increase the usability of this product. I believe that MorphoNet as explained in this revision makes for an excellent tool on the level of use by individual researchers for integrative multi-disciplinary research.

That being said, I continue to have serious concerns about MorphoNet as a service through which researchers will share data with each other and/or for public educational outreach. I will go into further detail below.

As a tool for end-user researchers to visualize morphological data with other lines of evidence, I think that MorphoNet has the potential to be very useful for doing integrative multi-disciplinary research. Making the source code open source and adding additional tools have largely addressed my concerns

here, since the digital infrastructure behind MorphoNet can now be transparently examined by the community.

We thank the referee for recognising our efforts and the relevance of MorphoNet as a tool for multidisciplinary scientific research. The remarks she/he raised in the first round of reviewing, helped us strengthen and consolidate MorphoNet .

The additional public datasets that have been added to MorphoNet are an interesting contribution. They certainly are enjoyable to peruse, and they expand the diversity of data available to users. They do inspire some questions that I think are relevant to consider. For one, I'm not sure why datasets were taken from MorphoMuseum and Phenome10k but not MorphoSource, since MorphoSource has thousands of datasets that are public access just like those from MorphoMuseum and Phenome10k. The authors state that "we contacted the curators of MorphoSource, but we have not been given access to their data." But MorphoSource has no centralized curators that determine access or lack thereof to data on that repository platform, as it is a platform designed to provide individual data uploaders the tools to control access to their own data. Since digital data is relatively novel, special care must be taken to allow for highly variable preferences by data contributors regarding the restricting of access and discovery. I'm not sure whether and how MorphoNet addresses this issue, but in any case there are many, many open access datasets on MorphoSource they could potentially use.

We understand the referee's concern, but we would like to stress that MorphoNet is presented here as a generic tool (akin to a genomic browser), which can be instantiated for a variety of projects by different communities. As such, while it is important to demonstrate that it can integrate a broad set of public datasets available in online repositories, it is difficult to foresee all possible uses. That said, and in response to the referee's suggestion, we now show as a further proof of principle that a computed tomography MorphoSource dataset, a Human bone dagger, can indeed be integrated into Morphonet.

Additionally, the practice of "reaccessioning" datasets from one digital 3D repository into another gives me pause. The ideal is of course for digital data to be easily available across distributed formats and sources. But if (for example) one repository encourages tracking of data usage and datasets from that repository are mirrored in a second repository that does not do similar tracking, does this dilute the quantitative evidence for data use and make users less likely to share digital data? There are likely ways to address this issue, but it has not been strongly addressed here. And I do recognize that this is not the "point" of MorphoNet per se, but I feel the reaccessioning datasets from other repositories raises some of these questions.

What the referee mentions here is a substantial, very general and complex issue for which there currently is, to our knowledge, no standard solution. Even a major resource on data and metadata standards, FAIRsharing, does not provide a solution for this. Solving the reaccessioning issue would need all major stakeholders to sit together and craft adequate guidelines. This is much beyond the scope of our work.

We do however share the concern of the referee, and decided to stress the issue in the manuscript. We have now included a couple of sentences in the main text acknowledging the issue, and reminding users that they should cite both our interactive visualization tool and the original data source.

These sentences are in lines 207 and read:

'Due to the lack of unified formats for morphological data and metadata, uploading a dataset from an external database in Morphonet leads to the duplication of this dataset in a slightly different format. To reduce the impact of this reaccessioning issue, we invite Morphonet users to always provide the original dataset ID in the metadata of the uploaded copy, and to cite both MorphoNet and the original database and ID in scientific publications'

Services for data sharing should ideally be built in such a way as to increase the traceability of data and to link it with both the physical specimens from which the digital data derives and any original or secondary sources from which digital data is gathered. MorphoNet does provide links per dataset to two sources: the source of digital data and a cited publication. This does provide an example of a way to trace data, and the sourcing is very clear. But MorphoNet does not seem to include in its data model any facility for explicitly categorizing datasets according to the physical objects from which

digital data derives. Rather, MorphoNet seems to organize its media into datasets, where each dataset can have some information regarding the biological specimen it represents. How does this model interact with the case where for a single specimen, multiple datasets have been produced? It seems these would be listed as two entirely separate datasets.

In MorphoNet, the source of original data and the corresponding publication the metadata associated to each dataset originally included a "description" field which could accommodate any desired information, including physical specimen information. We agree, however, that the specimen information is of a peculiar importance. To help tracing datasets originating from the same physical specimen, we have thus now introduced a specific "Physical specimen" metadata field. With this modification, categorizing datasets according to physical specimen would work well for datasets originating from databases such as Morphosource. In many other cases, however, for example in live imaging datasets, the physical specimen is not identified. Categorizing datasets according to the imaged dataset is thus more generally applicable than a categorization according to physical specimen.

Additionally, the higher level organization of datasets seems to be more or less arbitrary. One dataset is sorted into strictly taxonomic categories such as "Ascidian -> Phallusia mammillata", while another is sorted using taxonomic and topical categories such as "Human -> Anatomy", and yet another is sorted using taxonomic and paleontological categories such as "Fossil -> Crustacean". Certainly all of this metadata (and I do think that metadata is a more explicit and specifically recognized term than "information" and would strongly suggest the authors consider adopting it) should be tracked, but placing it all together in a single system of hierarchical categories divorced from context is only likely to confuse rather than to assist in data discovery and access. This is the basic gist I'm getting at: that the basic structure of MorphoNet makes it an excellent tool for a small number of datasets when used by a few individuals, but significant problems are likely to be encountered with any significant scaling.

We agree with what the referee is expressing here. In order to facilitate dataset exploration and sharing we have now implemented a taxonomy-based category system, thanks to which datasets can be searched, browsed and categorised according to the NCBI taxonomy database (<https://www.ncbi.nlm.nih.gov/Taxonomy/taxonomyhome.html/>). For this task, we have developed a dedicated API to import the NCBI taxonomy and label each dataset. In addition, each dataset is now labeled as: observed dataset, simulated dataset or drawing. These improvements have led to the addition of new classification menus and a search field at the top of the page on which all datasets are listed (<http://www.morphonet.crbm.cnrs.fr/listdatasets.php>). We believe these added features should facilitate scaling.

In conclusion, I feel that MorphoNet has significant potential to be extremely useful for a subset of the use cases that the authors have proposed for this tool, that being used by an individual for their own scientific research. Regarding other proposed use cases, such as a service for data sharing, data archiving, or as server-side software to be deployed by technically inclined end users, I feel that MorphoNet is of limited utility and requires significant further development and rethinking before it can serve those aims.

We thank the referee for her/his constructive suggestions. We hope that the upgraded version we now propose, and notably the new and much clearer structure of dataset organisation, will convince the referee that MorphoNet can and will be both a powerful tool for multidisciplinary research and an innovative and efficient system for data and metadata sharing across communities.

Morphonet is indeed not designed for data archiving, but we believe it will be used by many communities as a server-side software because of its power, because the code is open-source and versioned in INRIA's GitLab and because of the efforts we put into the writing of a detailed installation protocol accompanying the code (<https://gitlab.inria.fr/efaure/MorphoNet/blob/master/InstallMorphoNet.md>).

Reminder, access to MorphoNet : At the moment, access to the service requires users to register online. Once the work is published, public datasets will be accessible by any unregistered user. To facilitate the assessment of the tool, we have created a special referee account on MorphoNet, which gives reviewers the necessary rights to access a large part of the information stored on the server and to exploit the full functionality of MorphoNet:

url: <http://www.morphonet.crbm.cnrs.fr/>

login: reviewer.access

password: review

GitLab account : The access to this GitLab account is at the moment restricted to authenticated users only. It will be made open-access upon acceptance of the manuscript.

Address : <https://gitlab.inria.fr/efaure/MorphoNet> (click on **Standard** authentication)

Login : morphonet

Password : M0rph0\$e!